# AVTrack: Audio-Visual Tracking in Human-centric Complex Scenes

**Yaoting Wang**[1]  **Yun Zhou**[1]  **Zipei Zhang**[1]  **Henghui Ding**[1]

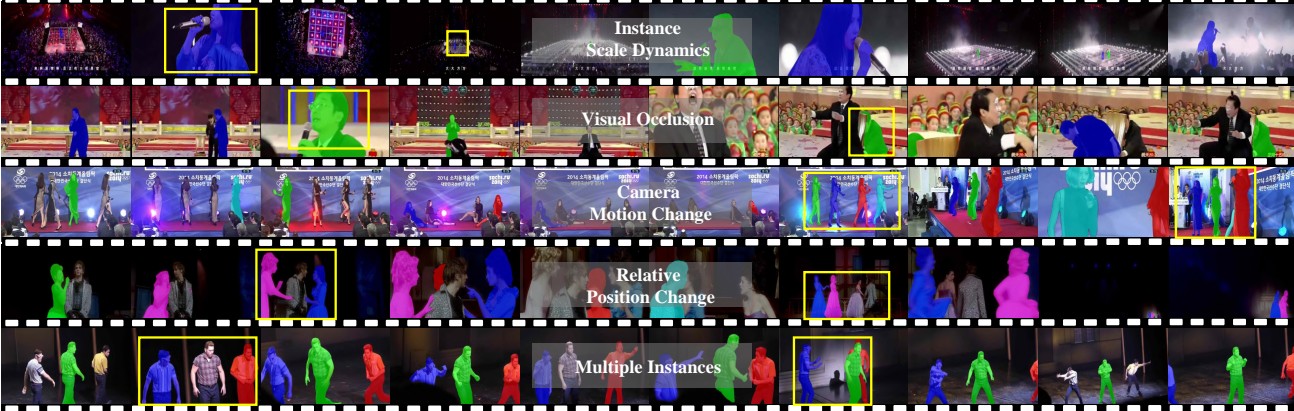

*Figure 1.* Illustrative samples from our proposed AVTrack benchmark. Audio signals are omitted for visual clarity. AVTrack features challenging human-centric audio-visual scenarios, such as instance scale dynamics, visual occlusion, camera motion change, and relative position change. In contrast, previous datasets are typically dominated by simple settings, such as static cameras and single-instance scenes. A more detailed description is provided in Section 3. Zoom in to inspect the instances in extremely small sizes.

## Abstract

Audio-visual speaker tracking aims to localize and track active speakers by leveraging auditory and visual cues, enabling fine-grained, human-centric scene understanding. This capability is essential for real-world applications such as intelligent video editing, surveillance, and human–computer interaction. However, existing datasets are largely limited to simple or homogeneous audio-visual scenes with coarse annotations. Such oversimplified settings bias evaluation toward static audio–visual co-occurrence, rather than rigorously assessing robust spatiotemporal modeling and cross-modal reasoning in complex, dynamic scenes. To address these limitations, we introduce **AVTrack**, a human-centric audio-visual instance segmentation (AVIS) dataset designed for dynamic real-world scenarios. AV-Track features diverse and challenging conditions, including camera motion, visual occlusions, and position changes. Evaluations of representative AVIS methods on AVTrack reveal substantial performance degradation, establishing AVTrack as a challenging benchmark for robust human-centric audio-visual scene understanding in complex environments. We further provide a simple yet effective baseline to facilitate future research. Project website: https://FudanCVL.github.io/AVTrack/.

## 1. Introduction

Humans naturally perceive and interpret their surroundings by seamlessly integrating visual and auditory signals. This cross-modal capability allows us to effortlessly identify who is speaking in a group, track a clapping individual amid visual clutter, or follow human activities under occlusion and rapid motion. Inspired by this perceptual ability, prior work (Li et al., 2022; Zhao et al., 2023; Li et al., 2025) has explored *audio-visual speaker tracking*, which leverages synchronized audio and visual streams to localize sounding individuals while maintaining identity consistency over time. As a core component of multimodal perception, audio-visual tracking underpins a wide range of applications, including human–computer interaction, augmented reality, security surveillance, and intelligent video editing.

Beyond speaker localization, recent progress in *audio-visual segmentation (AVS)* (Senocak et al., 2018; Zhou et al., 2022;

[1]Institute of Big Data, College of Computer Science and Artificial Intelligence, Fudan University, Shanghai, China. Correspondence to: Henghui Ding <hhding@fudan.edu.cn>.

*Proceedings of the 43rd International Conference on Machine Learning*, Seoul, South Korea. PMLR 306, 2026. Copyright 2026 by the author(s).

Wang et al., 2024c; Guo et al., 2025) has significantly advanced multimodal scene understanding by enabling pixel-level delineation of sounding objects. Among these efforts, *audio-visual instance segmentation (AVIS)* (Guo et al., 2025) represents a critical step forward, aiming to simultaneously detect, segment, and track sounding instances across video frames at the object level. By integrating instance-level reasoning with audio-visual correspondence, AVIS holds substantial promise for robust perception in complex, real-world settings, particularly in human-centric scenes.

Building reliable audio-visual correspondence is central to effective fine-grained multimodal learning. Early works such as AVS-Bench (Zhou et al., 2022; 2025) establish end-to-end CNN-based frameworks for jointly modeling visual and auditory cues. To better capture long-range temporal dependencies, AVSegFormer (Gao et al., 2024) introduces transformer-based architectures that significantly enhance spatiotemporal audio-visual reasoning. More recently, GAVS (Wang et al., 2024a) demonstrates strong data efficiency by integrating audio-visual correlation adapters into the visual foundation model SAM (Kirillov et al., 2023) via adapter tuning. In parallel, COMBO (Yang et al., 2024) advances cross-modal fusion by explicitly modeling bilateral audio-visual relations. Building upon these advances, AVISM (Guo et al., 2025) combines Mask2Former (Cheng et al., 2022) and VITA (Heo et al., 2022) extending audio-visual understanding to the instance level with window-based attention to achieve memory-efficient AVIS.

Despite notable progress, existing tasks and benchmarks still fall short in realistically evaluating AVIS systems. Early audio-visual speaker tracking datasets (Lathoud et al., 2004; Qian et al., 2019; 2022) are mainly collected in controlled laboratory settings, with few speakers and largely static scenes. Although AVA-ActiveSpeaker (Roth et al., 2020) increases speaker diversity, it is restricted to a single data source (TV series) and provides only bounding-box annotations without cross-frames identity consistency. Meanwhile, AVS benchmarks (Zhou et al., 2022; 2025) offer pixel-level labels in in-the-wild scenarios, but are dominated by short, trimmed clips (5–10 seconds), limiting long-range temporal modeling. AVISeg (Guo et al., 2025) partially addresses this issue by extending clip duration to around 60 seconds for instance-level segmentation; however, most scenes remain visually simple, with limited camera motion, background variation, and relative position shifts, particularly in human-centric scenarios. Such oversimplified settings fail to capture human-centric real-world challenges such as frequent occlusions, multi-person interactions, and complex spatiotemporal dynamics in the human-centric applications.

To address these challenges, we introduce **AVTrack**, a novel audio-visual benchmark designed for human-centric AVIS in complex scenes. AVTrack includes 871 video clips for 3,120 densely annotated instance tracklets, spanning 8 challenging conditions such as Camera Motion Change, Visual Occlusions, and Multiple Instances, which significantly complicate the modeling of cross-modal spatiotemporal dynamics. We benchmark state-of-the-art Video Instance Segmentation (VIS) and AVIS methods on AVTrack and observe significant performance degradation compared to existing datasets. These findings highlight AVTrack as a more realistic and challenging testbed for evaluating the robustness and advancements of human-centric AVIS systems. Additionally, we introduce a simple yet effective baseline framework to further support the community's research in fine-grained and human-centric audio-visual understanding.

In summary, our main contributions are as follows:

- We introduce **AVTrack** for benchmarking human-centric AVIS in complex and dynamic scenes.

- Experiments on AVTrack reveal significant performance gaps for VIS and AVIS methods, and we present a simple yet effective baseline with modular, plug-and-play extensibility for future research.

- We perform an in-depth analysis of the challenges posed by AVTrack and discuss promising directions for advancing human-centric AVIS in complex scenes.

## 2. Related Works

### 2.1. Audio-Visual Speaker Tracking

Audio-visual speaker tracking aims to track active speakers. Prior work has primarily relied on a small set of controlled datasets that limit real-world complexity. The most commonly used classic corporas such as AV16.3 (Lathoud et al., 2004) and CAV3D (Qian et al., 2019) provide multi-speaker audio-visual recordings in indoor meeting settings, but consist of recordings with controlled camera and microphone configurations, constraining diversity in scene dynamics and speaker variation. The AVRI (Qian et al., 2022) dataset increases recording duration, but remains constrained in environment and annotation scope. These benchmarks are valuable for evaluating tracking algorithms but do not capture the complexity of unconstrained real-world audio-visual scenarios beyond laboratory conditions.

### 2.2. Audio-Visual Segmentation

AVS (Zhou et al., 2022) aims to locate and segment sounding objects in a video, serving as a key step toward multimodal scene understanding. The pioneering work AVS-Bench (Zhou et al., 2022) establishes the foundational benchmark with audio-visual object segmentation for single sound sources (AVS-SS) and for multiple sound sources (AVS-MS), which focus on producing binary masks corre-

sponding to active audio emitters. Building upon this, audio-visual semantic segmentation (AVSS) (Zhou et al., 2025) further enriches the task by introducing semantic labeling, requiring the model to predict category-specific segmentation maps aligned with sound cues. Ref-AVS (Wang et al., 2024c) further advances multimodal grounding by introducing the reference audio-visual segmentation (Ref-AVS) task, which leverages textual cues describing visual and auditory information to guide flexible, context-aware segmentation. Recently, AVISeg (Guo et al., 2025) extends the audio-visual domain to AVIS, integrating detection, instance-level mask generation, and temporal association across frames. AVS is challenging due to complex cross-modal correspondence and multi-source interference. Existing AVS datasets, however, mostly contain simple and static scenes, limiting their ability to reflect real-world audio-visual complexity.

### 2.3. Complex Scene Understanding

Complex scene understanding has long been a challenge in computer vision, requiring robust reasoning under occlusion, dense interactions, and long-range temporal dependencies. Building on early work in cluttered environments, OVIS (Qi et al., 2022) establishes the first large-scale benchmark for occluded video instance segmentation, and MOSE (Ding et al., 2023) extends it to more realistic settings with dense crowds, camera motion, and large spatial displacements. Complex scene understanding has also been explored in audio-visual settings. AVS-MS (Zhou et al., 2022) introduces multi-source audio conditions, requiring the simultaneous segmentation of multiple sounding objects. More recent works (Wang et al., 2024b; Zha et al., 2025) emphasize the inherent inconsistency between auditory and visual cues, highlighting the need for coherent cross-modal reasoning beyond simple audio-conditioned segmentation. Despite these advances, existing AVS benchmarks (Zhou et al., 2022; 2025; Wang et al., 2024c) are largely limited to short clips of 5–10 seconds with simplified spatiotemporal structures, often reducing the task to frame-level audio-conditioned image segmentation. AVISeg (Guo et al., 2025) partially alleviates this limitation by extending video duration to around 60 seconds, yet it still lacks sufficient scene diversity and temporal complexity. In contrast, our work focuses on human-centric AVIS in 8 complex scenes, aiming to capture dynamic and intricate instance-level audio-visual interactions in real-world settings.

## 3. AVTrack Dataset

The AVTrack dataset is purpose-built to evaluate *human-centric AVIS* under realistic and highly challenging conditions. Unlike existing benchmarks that prioritize training supervision, AVTrack is released exclusively as a test set. By decoupling evaluation from dataset-specific train-

ing, this design choice reflects our primary objective: to establish a *rigorously curated, annotation-intensive benchmark* that serves as a stable and long-term reference for measuring progress in complex, fine-grained human-centric audio-visual understanding.

### 3.1. Define Complex Audio-Visual Scenes

To move beyond the simple and static scenarios that dominate existing audio-visual instance segmentation (AVIS) datasets, we explicitly define and enforce a set of criteria for *complex audio-visual scenes*. These criteria are grounded in a systematic analysis of real-world human-centric video content and are specifically designed to capture challenging conditions that expose the limitations of current models. A video is retained in AVTrack only if it exhibits one or more of the following characteristics:

- **Visual Occlusion:** Sounding individuals are partially occluded by other objects or people, leading to overlapping instances and ambiguous visual boundaries.

- **Relative Position Change:** Spatial ordering of instances changes over time (e.g., a man moves from left to right of others), necessitating persistent identity tracking rather than static localization.

- **Background Switch:** The scene undergoes substantial background changes, such as transitions between distinct environments, disrupting visual continuity.

- **Camera Motion Change:** Pronounced camera viewpoint variations, including zooming, panning, and shot transitions, alter instance scale, perspective, and visibility.

- **Multiple Instances:** Multiple instances of the same semantic class (i.e., humans) appear simultaneously, with only a subset actively producing sound, increasing the difficulty of audio-visual association.

- **Multi-turn Sounding:** Speaking turns alternate frequently among different individuals, causing the active audio-visual correspondence to shift over time.

- **Audio-Visual Inconsistency:** The auditory signal does not trivially align with the visually salient instance, such as off-screen speakers or background narration.

- **Instance Scale Dynamics:** Sounding individuals may appear at extremely large or small spatial scales and undergo substantial scale variation over time, hindering reliable instance detection and consistent audio-visual correlation.

### 3.2. Dataset Statistics

Table 1 summarizes representative benchmarks for audio-visual and visual-only detection and tracking, with a particular focus on *in-the-wild* datasets that provide instance-level annotations. Among these, **AVTrack**, AVA-ActiveSpeaker (Roth et al., 2020), and YouMVOS (Qi et al.,

*Table 1.* Comparison of non-laboratory datasets for audio-visual and visual-only tasks. Only VIS and AVIS provide instance-level annotations. AVTrack is designed specifically for human-centric AVIS evaluation. **Test**: proportion of test-set; **Length**: average video duration; **Anno.**: annotation granularity; **Audio**: whether audio is provided; **Track**: whether cross-frame instance identity is available.

| Task | Dataset | Videos | Test | Length | Domain | Anno. | Audio | Track | Publication |
|------|---------|--------|------|--------|--------|-------|-------|-------|-------------|
| ASD | AVA-ActiveSpeaker (Roth et al., 2020) | 262 | 41.6 | 529.0s | Human | bbox | ✓ | ✗ | [ICASSP'20] |
| AVL | VGG-SS (Chen et al., 2021) | 5,158 | 100.0 | 10.0s | Common | bbox | ✓ | ✗ | [CVPR'21] |
| AVOS | AVSBench-O (Zhou et al., 2022) | 5,356 | 15.0 | 5.0s | Common | mask | ✓ | ✗ | [ECCV'22] |
| AVSS | AVSBench-S (Zhou et al., 2025) | 12,356 | 20.7 | 7.8s | Common | mask | ✓ | ✗ | [IJCV'25] |
| Ref-AVS | RefAVS-Bench (Wang et al., 2024c) | 4,002 | 20.4 | 10.0s | Common | mask | ✓ | ✗ | [ECCV'24] |
| Ref-VOS | J-HMDB Sentences (Gavrilyuk et al., 2018) | 928 | 100.0 | 1.0s | Human | mask | ✗ | ✗ | [CVPR'18] |
| VIS | YouTube-VIS (Yang et al., 2019b) | 2,883 | 11.9 | 4.6s | Common | mask | ✗ | ✓ | [ICCV'19] |
| | OVIS (Qi et al., 2022) | 901 | 17.1 | 12.8s | Common | mask | ✗ | ✓ | [IJCV'22] |
| | YouMVOS (Wei et al., 2022) | 200 | 15.0 | 333.1s | Human | mask | ✗ | ✓ | [CVPR'22] |
| AVIS | AVISeg (Guo et al., 2025) | 926 | 22.1 | 61.4s | Common | mask | ✓ | ✓ | [CVPR'25] |
| | **AVTrack (ours)** | 871 | 100.0 | 54.0s | Human | mask | ✓ | ✓ | [ICML'26] |

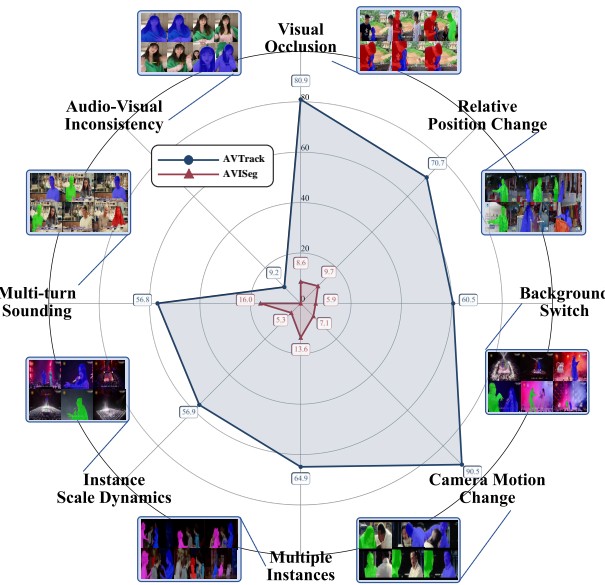

*Figure 2.* Comparison of AVTrack and AVISeg data distributions across different challenging conditions. Percentages indicate the proportion of each category relative to the total samples. The masked frames shown are sourced from AVTrack. **Zoom-in** for better visibility of details.

2022) are explicitly designed for human-centric understanding, while differing substantially in task formulation and supervision granularity. AVA-ActiveSpeaker targets active speaker detection (ASD) with frame-level binary speaking labels, YouMVOS emphasizes actor-centric multi-shot tracking, and OVIS addresses general object tracking in heavily occluded scenes without audio cues. In contrast, AV-Track is purpose-built for *human-centric AVIS* in dynamic audio-visual scenes, where complex acoustic conditions, diverse motion patterns, and rich scene composition jointly challenge spatio-temporal reasoning across modalities.

Figure 2 further contrasts the distribution of challenging scenarios between AVTrack and AVISeg. Compared to AVISeg,

AVTrack covers a substantially broader and more demanding spectrum of audio-visual conditions (see Section 3.1). In particular, AVTrack contains 9.2% Audio-Visual Inconsistency cases and exhibits pronounced increases in visual difficulty, including Visual Occlusion (80.9% vs. 8.6%), instance Scale Dynamics (56.9% vs. 35.3%), and Background Switch (60.5% vs. 5.9%). Beyond appearance-level challenges, AVTrack introduces significantly more complex temporal dynamics, such as Multi-turn Sounding (56.8% vs. 16.0%), Relative Position Change (70.7% vs. 9.7%), and Camera Motion Change (90.5% vs. 7.1%), collectively raising the bar for modeling long-range spatio-temporal variation and cross-modal alignment.

Another defining characteristic of AVTrack lies in the diversity of its data sources, as illustrated in Figure 3. Existing human-centric audio-visual datasets are often limited in scope: early speaker tracking benchmarks are predominantly captured in laboratory-controlled environments (Lathoud et al., 2004; Qian et al., 2019; 2022), while AVA-ActiveSpeaker (Roth et al., 2020) is largely derived from Hollywood movies. In contrast, AVTrack spans a wide range of video genres, including TV series, films, vlogs, animations, reality shows, interviews, and stage performances, while deliberately incorporating complex real-world scenarios. This breadth and heterogeneity more faithfully reflect real-world audio-visual conditions, enabling comprehensive and stress-tested evaluation of model robustness, generalization, and advanced cross-modal reasoning capabilities.

## 4. AVTracker: A Simple Baseline

### 4.1. Framework Overview

AVTracker serves as a simple yet effective modular multistage baseline for human-centric AVIS in complex audiovisual scenes. As shown in Figure 4, it adopts a divide-and-conquer strategy that first constructs audio-visual align-

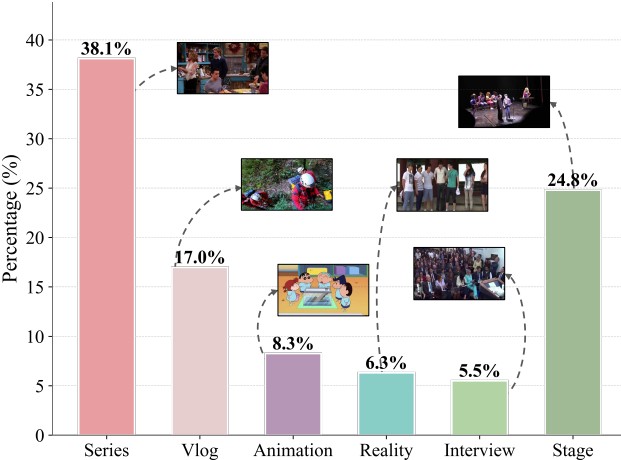

*Figure 3.* Video source distribution in AVTrack.

ments and local tracklets within temporally local windows, and then progressively associates tracklets of the same instance across windows to recover global speaker trajectories.

AVTracker is implemented as a modular three-stage framework explicitly designed for extensibility, allowing new tools and functional modules to be seamlessly integrated with minimal architectural modification. As a cascaded baseline system, AVTracker combines several advanced components to ensure strong performance across its audio-visual processing pipeline. Specifically, we adopt Qwen3-VL (Bai et al., 2025) as the visual reasoning backbone, Whisper (Radford et al., 2023) as the speech processor, and ECAPA-TDNN (Desplanques et al., 2020) as the speaker embedding encoder. To handle scenarios involving overlapping speech, MossFormer2 (Zhao et al., 2024) is optionally incorporated as a speech separation module. For spatial mask generation, we employ SAM3 (Carion et al., 2025) as the mask sampler. Following standard practice in audio-visual learning benchmarks (Zhou et al., 2022; Wang et al., 2024c; Guo et al., 2025), all videos are processed at a frame rate of $r = 1$ FPS. The speaker similarity threshold is set to $\tau = 0.35$.

### 4.2. Speaker Chunks Aggregation

A natural strategy for detecting speech activity is to leverage automatic speech recognition (ASR), which transcribes audio into semantically meaningful segments and enables AVTracker to operate on dynamically sized processing windows, as illustrated in Figure 5. Let $\mathcal{C} = \{c_i\}_{i=1}^N$ denote the ASR outputs including $N$ chunks, where each chunk $c_i = (t_i^s, t_i^e, x_i)$ comprises a start time, an end time, and a transcribed text. For each chunk, we apply a speech separation module $\mathcal{F}_{\text{sep}}$ to the corresponding raw audio segment $\mathbf{a}_i$, yielding an enhanced speech signal $\hat{\mathbf{a}}_i = \mathcal{F}_{\text{sep}}(\mathbf{a}_i)$.

In practice, ASR outputs often consist of short, fragmented, and temporally adjacent segments that may originate from

the same speaker. Processing such segments independently is inefficient and increases cost, hence we aggregate adjacent segments into longer speaker chunks to reduce the number of local windows and ease the burden on the global window. Specifically, given two consecutive chunks $c_i$ and $c_{i+1}$, we extract speaker embeddings using a pre-trained speaker encoder $\mathcal{E}$:

$$\mathbf{e}_i = \mathcal{E}(\hat{\mathbf{a}}_i), \quad \mathbf{e}_{i+1} = \mathcal{E}(\hat{\mathbf{a}}_{i+1}), \tag{1}$$

where $\hat{\mathbf{a}}_i$ denotes the corresponding audio segment. We then compute their cosine similarity:

$$\text{sim}(c_i, c_{i+1}) = \frac{\mathbf{e}_i^\top \mathbf{e}_{i+1}}{\|\mathbf{e}_i\|\|\mathbf{e}_{i+1}\|}. \tag{2}$$

If $\text{sim}(c_i, c_{i+1}) > \tau$, the two chunks are merged as

$$(t_i^s, t_i^e, x_i) \oplus (t_{i+1}^s, t_{i+1}^e, x_{i+1}) = (t_i^s, t_{i+1}^e, x_i \oplus x_{i+1}), \tag{3}$$

where $\oplus$ denotes temporal and textual concatenation. Then we obtain a set of speaker chunks $\mathcal{S} = \{s_k\}_{k=1}^K$.

### 4.3. Local Window Process

The local window process associates each speaker chunk with the visible person speaking within the corresponding temporal interval.

Given a speaker chunk $s_k = (t_k^s, t_k^e, x_k)$, we convert its temporal boundaries into video frame indices. The corresponding frame interval is computed as

$$f_k^s = \lfloor t_k^s \cdot r \rfloor, \quad f_k^e = \lfloor t_k^e \cdot r \rfloor. \tag{4}$$

For the frames $\{I_f\}_{f=f_k^s}^{f_k^e}$, where $I$ is a frame, we query the Local Reasoner $\mathcal{R}^{local}$ with the speech content $x_k$ and the visual observations to localize the active speaker with a frame-wise bounding box prediction $\mathbf{b}_{\mathcal{R}}^{(f)}$:

$$\mathbf{b}_{\mathcal{R}}^{(f)} = \mathcal{R}^{local}(I_f, x_k, P^{local}), \tag{5}$$

where $P^{local}$ is the prompt used in local processing. In parallel, SAM3 Video provides a set of person bounding boxes $\mathcal{B}_{\text{SAM3}}^{(f)}$ and corresponding masks $\mathcal{M}_{\text{SAM3}}^{(f)}$ for each frame. We align the $\mathcal{R}^{local}$ prediction with SAM3 detections by maximizing the intersection-over-union (IoU):

$$\mathbf{b}^{(f)} = \arg\max_{\mathbf{b} \in \mathcal{B}_{\text{SAM3}}^{(f)}} \text{IoU}\left(\mathbf{b}_{\mathcal{R}}^{(f)}, \mathbf{b}\right), \tag{6}$$

and retrieve the associated mask $\mathbf{m}^{(f)}$. The sequence of matched masks across the chunk forms a local tracklet:

$$\mathcal{T}_k^{\text{local}} = \{(f, \mathbf{m}^{(f)}) \mid f_k^s \leq f \leq f_k^e\}. \tag{7}$$

We further select a representative key frame as

$$f_k^{\text{key}} = \arg\max_{f \in [f_k^s, f_k^e]} \text{Area}\left(\mathbf{m}^{(f)}\right), \tag{8}$$

which is subsequently used for global identity association.

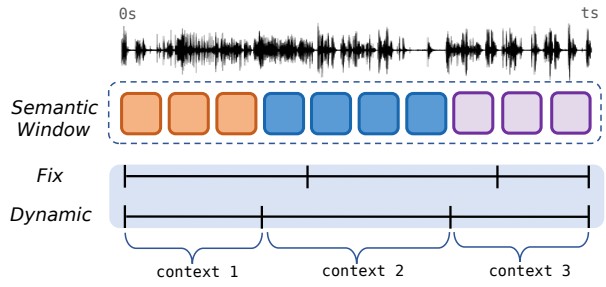

*Figure 4.* Overview of AVTracker, a three-stage framework for human-centric AVIS. **Stage 1 (Speaker Chunks Aggregation):** Speech clips are transcribed with Whisper, and timestamp-aligned transcripts together with speaker embeddings are used to group clips into speaker chunks, reducing redundancy and cost. Speech separation is ignored for clearer visualization. **Stage 2 (Local Window Process):** For each chunk, SAM3 segments candidate individuals in video frames. A Local Reasoner jointly conditions on speech transcripts and visual observations to associate utterances with visible persons, producing fine-grained, person-specific local tracklets. **Stage 3 (Global Window Process):** A Global Reasoner aggregates local tracklets across the entire video and resolves cross-chunk identity associations, yielding coherent instance-level speaker trajectories.

*Figure 5.* Comparison of windowing paradigms. Unlike fixed-size windows, dynamic windows preserve complete semantic units and contextual temporal continuity, enabling more effective and semantically coherent audio-visual correlation.

### 4.4. Global Window Process

While local windows resolve short-term audio-visual correspondence, the same speaker may appear in multiple disjoint chunks. The global window process aggregates local tracklets into consistent speaker identities. Let $\mathcal{F}_{\text{key}} = \{I_{f_k^{\text{key}}}\}_{k=1}^K$ denote all key frames. A Global Reasoner $\mathcal{R}^{global}$ is queried to group these frames according to person identity, producing a mapping:

$$\mathcal{G} : p \mapsto \{k_1, k_2, \dots\}, \qquad (9)$$

where each identity $p$ corresponds to a set of speaker chunks. For each identity $p$, we collect all associated local tracklets and merge them into a global trajectory:

$$\mathcal{T}_p = \bigcup_{k \in \mathcal{G}(p)} \mathcal{T}_k^{\text{local}}. \qquad (10)$$

For frames without observations, empty masks are inserted to maintain temporal continuity. The final output is a set of speaker trajectories $\{\mathcal{T}_p\}$ covering the entire video.

## 5. Experiments

### 5.1. Dataset Necessity and Challenges

As shown in Table 2, both VIS and AVIS methods experience pronounced performance degradation on the AV-Track benchmark. VIS-only approaches yield extremely low HOTA scores (below 12.0), indicating that visual cues alone are fundamentally insufficient for human-centric tracking in complex audio-visual scenarios. While AVIS methods incorporate audio information and approximately double the HOTA scores of VIS baselines (remaining below 21.0), their performance is still far from satisfactory. This reveals that existing audio-visual modeling paradigms struggle to achieve robust long-term identity association under the fragmented, asynchronous, and human-centric conditions inherent to AVTrack, corresponding to the eight challenges summarized in Section 3.1.

In contrast, AVTracker consistently outperforms all prior methods across all evaluation metrics, achieving improvements of approximately or exceeding 8.0 points over the strongest AVIS baseline. These gains highlight the critical importance of robust long-range identity association in complex audio-visual scenarios. AVTracker is designed as a modular and extensible framework, providing a strong and flexible baseline for future research on human-centric audio-visual understanding in complex scenes.

*Table 2.* Evaluation of VIS and AVIS methods on the AVTrack benchmark. AVIS models are trained on AVISeg, while VIS models are pretrained on YouTube-VIS and fine-tuned on AVISeg. All values are in percentage.

| Task | Method | HOTA ↑ | DetA ↑ | AssA ↑ | IDF1 ↑ | MOTA ↑ | Publication |
|---|---|---|---|---|---|---|---|
| VIS | VITA (Heo et al., 2022) | 9.70 | 10.54 | 9.35 | 12.32 | 1.91 | [NeurIPS'22] |
| | LBVQ (Fang et al., 2024) | 10.29 | 11.77 | 9.36 | 12.87 | 1.98 | [TCSVT'24] |
| | CAVIS (Lee et al., 2025) | 11.46 | 12.10 | 10.07 | 12.95 | 1.96 | [ICCV'25] |
| AVIS | AVISM (Guo et al., 2025) | 20.84 | 23.22 | 19.53 | 26.57 | 3.95 | [CVPR'25] |
| | ACVIS (Seo et al., 2025) | 20.60 | 22.59 | 19.66 | 26.23 | 4.23 | [ICASSP'26] |
| | AVTrackFormer (see Section D) | 21.47 | 22.51 | 20.26 | 26.41 | 4.11 | – |
| | **AVTracker** | **29.08** | **31.18** | **28.47** | **34.55** | **16.20** | [ICML'26] |

## 5.2. Ablation Study

AVTracker is designed with modularity and extensibility in mind, serving not only as a strong reference method but also as a flexible research baseline for future studies on human-centric audio-visual understanding in complex real-world scenes. As shown in Table 3, we conduct ablation studies to examine the contribution of different components in the proposed AVTracker framework. Specifically, we analyze the effects of (i) the model scale, (ii) the speech separation and (iii) the chunks processing.

**Effect of Model Scale.** We examine the impact of model scale in both the speech processor and the VLM reasoner. Compared with the `Base` setting, reducing the capacity of either the speech processor (`M1`) or the VLM (`M2`) results in notable performance degradation, indicating that sufficient representational capacity in both modalities is critical for effective audio-visual reasoning. When further reducing both the speech processor and VLM capacity in `M3`, performance drops become more pronounced, with HOTA and AssA decreasing by 4.84 and 3.70 points, respectively. We additionally replace the VLM-based global reasoner with face detection features (Serengil & Özpınar, 2024) for local tracklet grouping, which leads to even larger declines on HOTA (−5.23) and AssA (−6.08). These results show that strong audio-visual back end is essential for robust long-range identity association.

**Effect of Speech Separation.** We analyze the role of speech separation for tackling potential overlapping conditions. Compared with the `Base` setting, incorporating Moss-Former (Zhao et al., 2024) for speech separation (`S2`) leads to improvements in both HOTA (28.85 → 29.08) and AssA (27.39 → 28.47). In contrast, applying with SepFormer (Wu et al., 2022) (`S1`) slightly degrades performance (HOTA: 28.85 → 28.41), showing that imperfect separation may introduce additional noise and temporal misalignment. These results highlight that speech separation can be beneficial, but only when the separation quality is sufficiently reliable to support downstream audio-visual alignment.

**Impact of Chunk Processing.** Removing local chunk compression (`C1`) leads to a severe performance drop compared

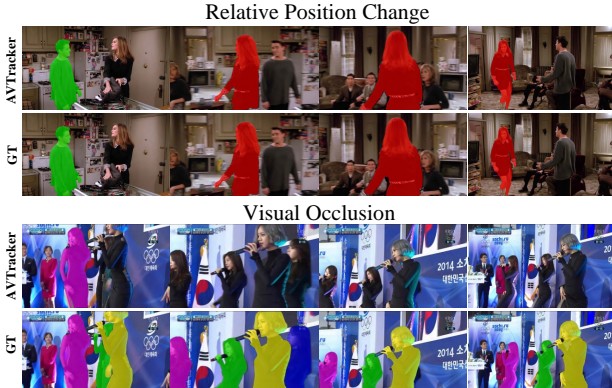

*Figure 6.* Qualitative comparison between AVTracker predictions and ground truth labels.

with `M3`, with HOTA decreasing from 24.01 to 16.88, showing the importance of compact and informative local tracklets for scalable global association. In addition, disabling dynamic windowing (`C2`) also results in a performance decline relative to the `Base` setting (HOTA: 28.85 → 27.45).

## 5.3. Qualitative Comparison

Figure 6 shows a qualitative comparison between AVTracker predictions and ground-truth labels. AVTracker tracks the active speaker well under relative position changes but struggles when multiple speakers and visual occlusions occur together. More visualizations are in Section E.

## 6. Discussion and Future Directions

Based on the challenges posed by AVTrack, we outline several directions for future research in human-centric audio-visual scene understanding.

**Robust Audio-Visual Alignment.** Scenarios involving multi-turn speech, visual occlusion, audio-visual inconsistencies, and multiple active speakers present extreme challenges for cross-modal alignment. A key factor in AV-Tracker's effectiveness is its use of textual cues as a semantic bridge between audio and visual streams. In addition, its dynamic windowing mechanism allows complete utterance-

*Table 3.* Ablation study of different model configurations on AVTrack benchmark. 244M and 809M `Param`$_A$ corresponds to Whisper-small and Whisper-large-v3-turbo, respectively, while 4B and 8B `Param`$_{VL}$ correspondes to Qwen3-VL-4B-Instruct and Qwen3-VL-8B-Instruct. **Cmpr.:** whether local speech chunks are compressed to reduce computational load in the global window; **Chunk**: whether a dynamic windowing strategy is used for chunking; **Sepa.:** whether speech separation is applied to handle overlapping speech.

| Setting | Param$_A$ | Param$_{VL}$ | Cmpr. | Chunk | Sepa. | HOTA ↑ | DetA ↑ | AssA ↑ | IDF1 ↑ | MOTA ↑ |
|---|---|---|---|---|---|---|---|---|---|---|
| | | | | | AVTRACKER BASE SETTING | | | | | |
| Base | 809M | 8B | ✓ | ✓ | ✗ | 28.85 | **31.75** | 27.39 | 34.45 | **16.39** |
| | | | | | IMPACT OF MODEL SIZE | | | | | |
| M1 | 244M | 8B | ✓ | ✓ | ✗ | 25.19 | 27.33 | 24.25 | 29.92 | 14.88 |
| M2 | 809M | 4B | ✓ | ✓ | ✗ | 24.47 | 25.85 | 24.37 | 28.86 | 14.48 |
| M3 | 244M | 4B | ✓ | ✓ | ✗ | 24.01 | 25.49 | 23.69 | 28.47 | 13.52 |
| M4 | 244M | 4B/Face | ✓ | ✓ | ✗ | 23.62 | 24.80 | 21.31 | 27.16 | 11.03 |
| | | | | | IMPACT OF SEPARATION | | | | | |
| S1 | 809M | 8B | ✓ | ✓ | SepFormer | 28.41 | 30.81 | 27.54 | 33.65 | 15.99 |
| S2 | 809M | 8B | ✓ | ✓ | MossFormer2 | **29.08** | 31.18 | **28.47** | **34.55** | 16.20 |
| | | | | | IMPACT OF CHUNK PROCESSING | | | | | |
| C1 | 244M | 4B | ✗ | ✓ | ✗ | 16.88 | 18.34 | 16.33 | 19.99 | 9.34 |
| C2 | 809M | 8B | ✓ | ✗ | ✗ | 27.45 | 29.57 | 26.64 | 32.97 | 13.49 |

level semantic information to be captured by the visual reasoning module. Future work could explore methods that further improve alignment under such complex conditions.

**Audio-Visual Spatio-Temporal Intelligence.** Dynamic changes in camera motion, relative positions, and instance scales introduce additional challenges for audio-visual cross-modal reasoning. AVTracker leverages the reasoning ability of VLM without task-specific training and carefully tuned prompts, achieving substantial performance improvements. Future research may investigate ways to enhance spatio-temporal reasoning capabilities to better handle these complex dynamics.

**Efficient Human-centric Data Construction.** AVTrack is designed as a carefully constructed, test-only benchmark to provide a long-term, stable evaluation of human-centric audio-visual scene understanding. It remains an open question whether scaling up training data can help address the challenges exposed by AVTrack. Future work could focus on building efficient pipelines for human-centric data collection, potentially structured around *person–time–location* triplets, and explore whether training-based approaches can bridge the observed performance gaps.

**Agentic Audio-Visual Reasoning.** The challenges exposed by AVTrack, such as fragmented observations, long-term speaker identity maintenance, and ambiguous audio-visual cues, naturally motivate agentic reasoning formulations for human-centric audio-visual understanding. Rather than relying solely on one-step reasoning, future work could investigate agentic reasoning (Wei et al., 2026) that iteratively reason over scenes through memory (Zhang et al., 2025; Xu et al., 2025) and reflection (Shinn et al., 2023). Explicit memory mechanisms may allow agents to accumulate evidence across temporal windows and recover from tran-

sient alignment errors, while reflection or self-correction (Pan et al., 2023; Gou et al., 2023) strategies could help revise earlier decisions when confronted with conflicting cross-modal signals. In addition, structured intermediate representations, such as dynamic scene graphs (Li et al., 2024; Fei et al., 2024) or speaker-centric relational graphs (VS et al., 2023), may provide a persistent abstraction for reasoning about entities and their interactions over time.

## 7. Conclusion

This work investigates fine-grained, human-centric audio-visual understanding and highlights the substantial gap between existing benchmarks and the complexity of real-world scenarios. To probe the performance limits of models in realistic human-centric audio-visual environments, we introduce **AVTrack**, a challenging dataset for human-centric audio-visual instance segmentation and tracking. AVTrack spans 6 common audio-visual data sources and covers 8 representative challenges, including visual occlusions, relative position changes, and significant camera motion. Through comprehensive evaluations, we observe that representative AVIS methods suffer from pronounced performance degradation on AVTrack, indicating that models trained on relatively static and simplified benchmarks struggle to generalize to complex real-world settings. To further advance this line of research, we propose a simple, effective, and extensible baseline that performs local-to-global tracklet grouping, demonstrating the feasibility of structured audio-visual reasoning and long-term tracking in challenging scenarios. We expect AVTrack to foster research on robust, human-centric audio-visual scene understanding and to inspire the development of next-generation models capable of fine-grained spatiotemporal reasoning in real-world environments.

## Acknowledgements

This work was supported by the Science and Technology Commission of Shanghai Municipality under Grant No. 25511103600 and the National Natural Science Foundation of China (NSFC) under Grant No. 62472104.

## Impact Statement

This work advances human-centric audio-visual tracking and scene understanding by introducing a challenging benchmark dataset and a baseline method. The proposed AVTrack aims to **support research** on robust audio-visual understanding in realistic, dynamic environments, with potential applications in video editing and human–computer interaction. However, technologies that localize and track speaking individuals raise privacy, surveillance, and misuse concerns, especially without proper safeguards. Although AVTrack uses only publicly available videos and is **intended strictly for research**, models trained on it could be repurposed in ways that compromise privacy or enable intrusive monitoring. We stress that **AVTrack is released solely as a research benchmark** to study current model limitations in complex audio-visual settings, not to enable real-world surveillance. We encourage future work to adopt **privacy-preserving** learning, responsible data practices, and ethical deployment. By highlighting challenges and limitations, *we aim to foster more transparent, accountable, and socially responsible human-centric perception systems*.

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

# A. Additional Related Works

## A.1. Video Instance Segmentation

Video instance segmentation (VIS) (Yang et al., 2019a) is the task of simultaneously detecting, segmenting, and tracking instances across frames in a video. Early VIS methods (Yang et al., 2019a; Cao et al., 2020; Athar et al., 2020; Liu et al., 2021a; Yang et al., 2021) mainly evolve from CNN-based image instance segmentation frameworks (He et al., 2017; Guo et al., 2021), incorporating temporal cues via optical flow, feature propagation, or tracking modules. For example, MaskTrack R-CNN (Yang et al., 2019a) extends Mask R-CNN (He et al., 2017) with an association branch for temporal linking, while SGNet (Liu et al., 2021a) adopts an anchor-free centerness-based design for robust correspondence. Transformer-based architectures then bring a paradigm shift toward end-to-end VIS. VisTR (Wang et al., 2021) first extends DETR (Carion et al., 2020) into a query-based formulation that directly predicts instance mask sequences, removing explicit association. Subsequent models like SeqFormer (Wu et al., 2022) and Mask2Former-VIS (Cheng et al., 2022) enhance query interaction and multi-scale aggregation, achieving stronger segmentation and tracking. Token-centric frameworks such as IFC (Hwang et al., 2021) and VITA (Heo et al., 2022) further improve efficiency and scalability by compressing spatio-temporal features into compact instance-aware tokens, enabling efficient long-range reasoning across videos. CAVIS (Lee et al., 2025) introduces a context-aware approach to VIS, enhancing object tracking and segmentation accuracy by incorporating surrounding contextual information across frames.

## A.2. Human-centric Object Tracking

Several human-centric video understanding datasets have been proposed to study people, actions, and interactions in complex real-world videos. SoccerNet (Giancola et al., 2018) introduces large-scale broadcast soccer videos for action spotting, focusing on temporally localized human actions and game events in long, untrimmed sequences. SoccerNet-v2 (Deliege et al., 2021) extends this line of work toward a more holistic understanding of soccer broadcasts by incorporating richer annotations and multiple complementary tasks, encouraging joint modeling of human actions, camera dynamics, and high-level semantics. Building further on human-centered analysis, SoccerNet-Tracking (Cioppa et al., 2022) shifts the focus to multiple object tracking, emphasizing consistent identity modeling of players, referees, and other key actors over time in crowded and dynamic scenes. Outside the sports domain, YouMVOS (Wei et al., 2022) addresses actor-centric video object segmentation in unconstrained videos, highlighting the challenge of maintaining coherent actor-centric instance

segmentation across multiple shots and scene changes. Together, these works reflect a progression from sparse action recognition toward comprehensive, human-centric understanding of long and complex videos.

## A.3. Person Re-identification (Re-ID)

Person Re-ID seeks to match images of the same individual across views or video shots, facing challenges from illumination, viewpoint, pose variations, occlusions, and background clutter. Traditional approaches focus on learning discriminative visual features, using attention mechanisms (He et al., 2021) to reduce background noise, part-based or locally aligned representations (Zhao et al., 2017; Li et al., 2021) to handle pose misalignment, and metric learning (Yi et al., 2014) to structure identity embeddings. The advent of VLMs has inspired integrating multimodal semantic information into ReID. VLMs pretrained on image–text pairs can capture high-level attributes such as clothing, accessories, or demographic cues, complementing visual features for open-vocabulary retrieval or text-based search. LLaVa-ReID (Lu et al., 2025) iteratively queries fine-grained attributes to refine ambiguous descriptions. TVI-LFM (Hu et al., 2024) leverages VLM- and LLM-generated text for joint modality alignment, improving infrared features and cross-modal consistency. MLLM4Text-ReID (Tan et al., 2024) automatically generates diverse textual templates and enhances cross-dataset transfer without target-domain fine-tuning. MLLMReID (Yang & Zhang, 2024) adapts visual encoders through a unified instruction strategy and synchronized multi-task training. This line of work suggests that leveraging high-level semantic reasoning improves robustness and generalization in challenging ReID settings, paralleling our emphasis on human-centric tracking.

# B. Data Collection and Annotation Pipeline

As shown in Figure 7, AVTrack is built through a carefully designed pipeline consisting of four key stages:

**Video Collection and Curation.** We first collect approximately 1,300 candidate videos from publicly available web sources, targeting real-world human-centric scenarios such as interviews and performances. These videos are subjected to an initial screening stage that removes samples failing to meet minimum requirements on the scene complexity mentioned above. After filtering, roughly 1,000 videos are retained as candidates for annotation. This curation step ensures that the dataset consistently reflects non-trivial audio-visual interactions beyond simple and static environments.

**Automatic Pre-annotation.** To bootstrap instance-level annotations at scale, we employ Grounded-SAM (Ren et al., 2024) to generate frame-level human instance masks for each candidate video. This automatic stage provides coarse

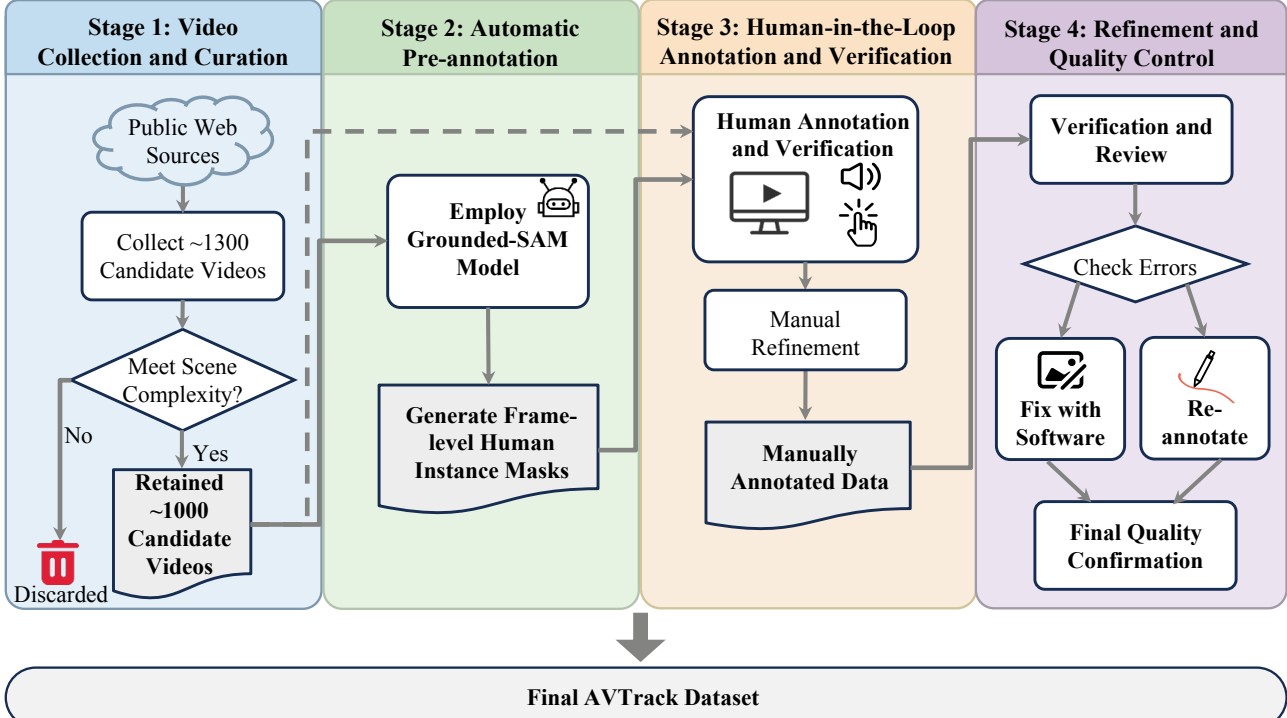

*Figure 7.* AVTrack dataset construction pipeline.

yet comprehensive spatial coverage and significantly reduces the manual annotation burden, while preserving the flexibility required for subsequent human refinement.

**Human-in-the-Loop Annotation and Verification.** To ensure annotation consistency and correctness, all annotators undergo structured training on a held-out subset of the data. Each annotator is required to complete an initial annotation pass, which is reviewed by the project team before full-scale labeling begins. We further develop a dedicated web-based annotation tool that enables annotators to track individual instances across frames and manually add or correct masks missed by the automatic process. During annotation, synchronized video, audio, and full-frame visual context are provided, and annotators are instructed to label *sounding human instances* on a per-frame basis.

**Refinement and Quality Control.** Following manual annotation, each video undergoes a refinement stage to correct residual errors. Minor artifacts, such as fragmented regions or small holes, are fixed using graphic editing software, while severely inaccurate or missing instances are re-annotated using LabelMe [1]. This multi-level verification process ensures spatial accuracy, temporal consistency, and reliable audio-visual correspondence across frames.

---

[1]LabelMe: https://github.com/wkentaro/labelme

**Final Dataset.** After rigorous quality control, the final AVTrack benchmark comprises **871 high-quality video clips** encompassing 3,120 annotated instances. Over a period of nearly three months, 15 professional annotators contributed to data collection, annotation, and verification. This meticulous pipeline resulted in a test-only dataset whose annotation fidelity and scene complexity surpass those of existing human-centric audio-visual benchmarks, providing a reliable and challenging evaluation platform for advancing robust audio-visual perception and long-term identity association in complex real-world scenarios.

## C. Evaluation Metrics

We evaluate human-centric AVIS using the TrackEval toolkit (Luiten et al., 2021), which assesses detection accuracy, temporal association, and identity consistency. These capabilities are critical in human-centric scenarios involving frequent speaker switches, occlusions, and long-range interactions. Following prior work (Cheng et al., 2022; Heo et al., 2022; Guo et al., 2025), we adopt five standard evaluation metrics: HOTA (Higher-Order Tracking Accuracy), DetA (Detection Accuracy), AssA (Association Accuracy), IDF1 (Identity F1 Score), and MOTA (Multiple Object Tracking Accuracy). All metrics are computed using the official API to ensure reproducibility and fair comparison, reported per video, and averaged across the dataset.

**HOTA.** Higher Order Tracking Accuracy (HOTA) jointly measures detection and association performance by averaging scores over different matching thresholds. Formally, HOTA is defined as

$$\text{HOTA} = \frac{1}{|\mathcal{A}|} \sum_{\alpha \in \mathcal{A}} \sqrt{\text{DetA}_\alpha \cdot \text{AssA}_\alpha}, \qquad (11)$$

where $\alpha$ denotes the matching threshold, and $\mathcal{A}$ is the set of thresholds used for evaluation. By explicitly balancing detection and association terms, HOTA provides a holistic assessment of tracking quality.

**Detection Accuracy (DetA).** Detection Accuracy evaluates frame-level instance detection correctness and is defined as

$$\text{DetA} = \frac{\text{TP}}{\text{TP} + \text{FN} + \text{FP}}, \qquad (12)$$

where TP, FN, and FP denote the numbers of true positives, false negatives, and false positives, respectively, based on the matching criterion used for evaluation. In the context of AVIS, DetA measures whether sounding human instances are correctly localized at the frame level.

**Association Accuracy (AssA).** Association Accuracy evaluates the correctness of temporal associations between matched detections across frames. It is defined as

$$\text{AssA} = \frac{1}{|\mathcal{C}|} \sum_{c \in \mathcal{C}} \frac{|\text{TPA}(c)|}{|\text{TPA}(c)| + |\text{FNA}(c)| + |\text{FPA}(c)|}, \qquad (13)$$

where $\mathcal{C}$ denotes the set of true positive detection matches obtained under the adopted matching criterion, and $\text{TPA}(c)$, $\text{FNA}(c)$, and $\text{FPA}(c)$ represent the numbers of true positive, false negative, and false positive associations induced by the matched detection $c$, respectively. AssA quantifies local association correctness and penalizes inconsistent temporal linkages, reflecting a model's ability to maintain coherent instance associations over time.

**IDF1.** IDF1 evaluates global identity preservation and is defined as

$$\text{IDF1} = \frac{2 \cdot \text{IDTP}}{2 \cdot \text{IDTP} + \text{IDFP} + \text{IDFN}}, \qquad (14)$$

where IDTP, IDFP, and IDFN denote the numbers of identity true positives, false positives, and false negatives, respectively. IDF1 is sensitive to long-term identity fragmentation and is particularly relevant for challenging dynamic human-centric scenes.

**MOTA.** Multi-Object Tracking Accuracy (MOTA) aggregates detection and identity errors into a single metric:

$$\text{MOTA} = 1 - \frac{\text{FN} + \text{FP} + \text{IDSW}}{\text{GT}}, \qquad (15)$$

where IDSW is the number of identity switches and GT denotes the total number of ground-truth instances.

## D. AVTrackFormer: An End-to-end Baseline

Upon observing AVISM's performance on AVTrack, we conducted a preliminary experiment by making minor modifications to AVISM, yielding a new variant named AVTrackFormer. Unlike AVTracker, AVTrackFormer is an end-to-end model trained in the same manner as AVISM (Guo et al., 2025) but features enhanced modeling of audio-visual interaction during the video-level sounding object tracking.

### D.1. Audio-Visual Representation

Given an input video with temporally aligned visual and audio streams, we segment it into $T$ consecutive, non-overlapping temporal snippets $\{(v_i, a_i)\}_{i=1}^T$, each spanning one second. For each visual snippet $v_i$, a visual backbone extracts multi-scale feature maps $f_{i,k}^V \in \mathbb{R}^{H_k \times W_k \times D_k}$, where $k$ indexes the stages of the backbone. The aggregated visual representation across all snippets is denoted as $F_V = \{f_i^V\}_{i=1}^T$. For each audio snippet $a_i$, we compute a log-mel spectrogram, which is then encoded into a fixed-dimensional embedding $f_i^A \in \mathbb{R}^D$ using a pre-trained audio encoder. The resulting audio representation for the video is denoted as $F_A = \{f_i^A\}_{i=1}^T$. The parameters of the audio encoder are kept frozen during training.

### D.2. Frame-Level Sound Source Localizer

We perform frame-wise localization of sounding objects by explicitly modeling spatial audio-visual correspondence. Given the visual features $f_i^V$, a pixel decoder generates enhanced visual representations $\hat{f}_i^V$ as well as per-pixel embeddings $p_i \in \mathbb{R}^{H \times W \times C}$. To integrate audio information, the Audio-Visual Pixel-level Fusion Module (AV-PFM) applies cross-attention between $\hat{f}_i^V$ and the corresponding audio embedding $f_i^A$, producing audio-conditioned visual features $f_i^M \in \mathbb{R}^C$. Following the set prediction paradigm (Carion et al., 2020; Guo et al., 2025), we introduce $N_f$ learnable frame queries $Q \in \mathbb{R}^{N_f \times C}$, each conditioned on the multimodal feature $f_i^M$, resulting in frame queries $Q_f \in \mathbb{R}^{N_f \times C}$. These queries are further refined through a Transformer decoder. Subsequently, each query is projected into the pixel embedding space via a dot product with $p_i$ to predict both the class label and segmentation mask corresponding to each sounding object.

### D.3. Video-Level Sounding Object Tracker

This module associates frame-level predictions across an entire video. In line with AVISM and to mitigate computational overhead in long or high-resolution videos, the tracker operates on frame queries rather than dense pixel features. Concretely, a linear layer projects the $T \times N_f$ frame queries from all frames into object tokens $Q_o$. Following VITA (Heo et al., 2022), these object tokens are processed

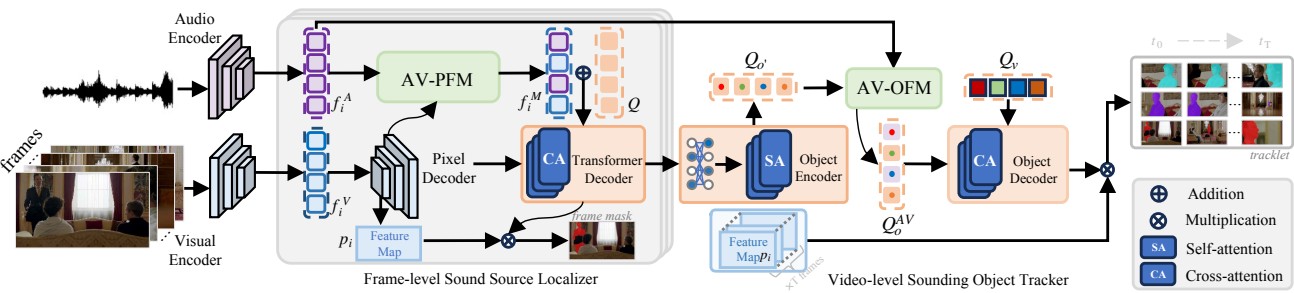

*Figure 8.* An overview of AVTrackFormer: similar to AVISM, it first performs pixel-level cross-modal fusion through AV-PFM. However, AVTrackFormer enables bidirectional interaction between object tokens and audio features in AV-OFM, rather than a unidirectional one.

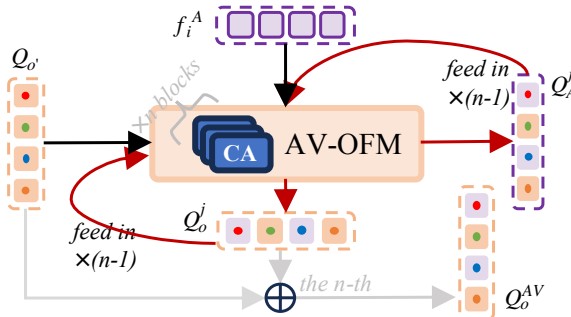

*Figure 9.* Audio-Visual Object-level Fusion Module (AV-OFM). AV-OFM is designed to model audio–visual correlations in an interleaved and bidirectional manner.

by an object encoder using windowed self-attention (Liu et al., 2021b), producing encoded tokens $Q_{o'}$. Audio cues are incorporated into the temporal modeling via the Audio-Visual Object-level Fusion Module (AV-OFM) applied to $Q_{o'}$, resulting in audio-conditioned object tokens $Q_o^{AV}$. To extract object-centric representations from all tokens, we initialize a fixed set of learnable video queries $Q_v \in \mathbb{R}^{N_v \times C}$, where $N_v$ denotes the number of video queries. A transformer decoder serves as the object decoder, which takes $Q_o^{AV}$ as input and aggregates their semantic information into the video queries. The decoder output are then dot-multiplied with $p_i$ to get the final mask logits.

**AV-OFM:** In contrast to AVISM, where cross-modal interaction is limited to a unidirectional enhancement, specifically, audio features $f_i^A$ only attend to object tokens $Q_{o'}$ to update themselves before being fused back, AVTrackFormer introduces the enhanced AV-OFM to enable bidirectional interactions. As illustrated in Figure 9, our AV-OFM treats object tokens $Q_{o'}$ and audio features $f_i^A$ as co-evolving states. We denote the intermediate states at the $j$-th block as $Q_o^j$ and $Q_A^j$, with initial states $Q_o^0 = Q_{o'}$ and $Q_A^0 = f_i^A$. For each iteration $j \in \{1, \ldots, n-1\}$, the module recursively updates both modalities:

$$(Q_o^j, Q_A^j) = \text{AV-OFM}(Q_o^{j-1}, Q_A^{j-1}). \quad (16)$$

*Table 4.* Impact of visual backbone on AVTrack. All models are evaluated with backbone pretrained on ImageNet and Coco dataset.

| Backbone | Model | HOTA | DetA | AssA | IDF1 | MOTA |
|---|---|---|---|---|---|---|
| R-50 | AVISM | 16.56 | 17.39 | 16.48 | 21.59 | 1.89 |
| | ACVIS | 16.94 | 18.02 | 16.62 | 22.55 | 2.18 |
| | AVTrackFormer | 17.50 | 18.91 | 16.97 | 22.78 | 2.25 |
| Swin-L | AVISM | 20.84 | **23.22** | 19.53 | **26.57** | 3.95 |
| | ACVIS | 20.60 | 22.59 | 19.66 | 26.23 | **4.23** |
| | AVTrackFormer | **21.47** | 22.51 | **20.26** | 26.41 | 4.11 |

Within this iterative process, the cross-attention mechanism (CA) facilitates reciprocal information exchange, where object tokens are conditioned on audio context and audio features are simultaneously refined using object-level visual cues. After the final $n$-th block, the output $Q_o^n$ is combined with the original input $Q_{o'}$ via a residual connection to yield the final fused representation:

$$Q_o^{AV} = Q_{o'} \oplus Q_o^n. \quad (17)$$

### D.4. Impact of Different Visual Backbone

Table 4 analyzes the impact of different visual backbones on AVTrack and representative AVIS models. Overall, replacing the ResNet-50 backbone with the stronger Swin-Large consistently leads to substantial performance improvements across all evaluated metrics and models, highlighting the critical role of high-capacity visual representations in complex audio-visual tracking scenarios.

Under the ResNet-50 backbone, all methods exhibit relatively limited performance, with AVTrackFormer achieving the best overall results among the three models, Indicating that more effective audio-visual interaction design can yield consistent gains in both detection quality and identity preservation. When adopting the Swin-Large backbone, the performance gap between methods becomes more nuanced. All models benefit notably from stronger visual features, with absolute improvements of approximately 3-4 points in HOTA compared to ResNet-50. AVTrackFormer attains the highest HOTA and AssA scores, suggesting superior association capability when richer visual context is available,

while AVISM and ACVIS achieve competitive results in DetA and MOTA, respectively. These results suggest that stronger backbones primarily enhance detection robustness, whereas improvements in association accuracy depend more

heavily on the tracking and fusion design.

## E. Qualitative Comparison

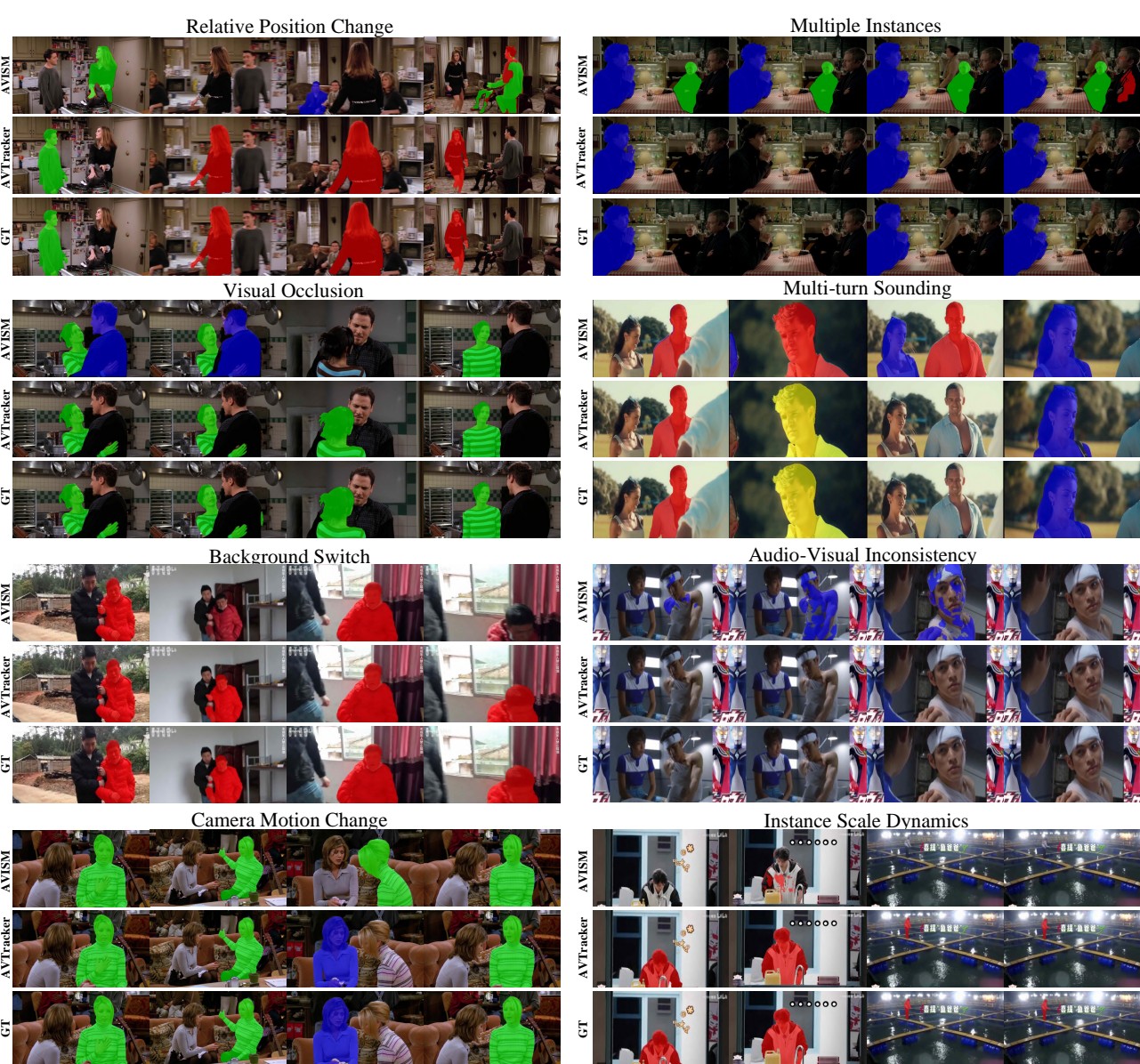

*Figure 10.* Qualitative comparison between AVTracker and AVISM.

As shown in Figure 10, in this section, we provides qualitative comparison across AVTracker and AVISM under different challenging cases.

In the **Relative Position Change** scenario, changes in relative positioning between speakers in the first and last frames may lead AVISM to mistakenly treat different individuals as the same. In the **Camera Motion Change** scenario, AVISM has limited robustness to camera motion, potentially resulting in incorrect audio-visual associations. In the

**Multi-turn Sounding** scenario, AVISM struggles with modeling multiple rounds of speech, often erroneously assigning the same instance ID to different speakers, which causes identity shifts. In contrast, AVTracker's modular "segment-then-match" design, coupled with a local-to-global tracklet grouping mechanism, supports accurate speaker identity assignment, handling these challenges more effectively.

In the **Visual Occlusion** scenario, AVISM demonstrates weak modeling of instance integrity when occlusion occurs.

For example, it may incorrectly assign a woman's hand to a man in the first and second frames. In the **Background Switch** scenario, AVISM is sensitive to background changes, which leads to unstable speaker detection. In the **Instance Scale Dynamics** scenario, AVISM has limited performance for small-scale instances, leading to suboptimal detection and tracking of distant or small targets. In the **Multiple Instances** scenario, AVISM struggles to establish precise audio-visual correspondences between individuals with similar appearances or those located close to each other, resulting in instance confusion. In contrast, AVTracker effectively handles these issues, leveraging a robust pre-trained segmentation backbone to enhance performance.

In the **Audio-Visual Inconsistency** scenario, AVISM has weak joint audio-visual understanding, occasionally attributing speech to visually salient but silent objects. On the other hand, AVTracker employs dynamic windowing to integrate semantic information, effectively linking speech semantics with visual cues and accurately resolving audio-visual inconsistencies.

## F. Implementation Details

### F.1. Model Configuration

AVTracker is composed entirely of off-the-shelf pretrained models, which makes the pipeline easy to reproduce and to upgrade as stronger components become available. The specific variants used in all reported experiments are listed in Table 5.

*Table 5.* Model configuration of AVTracker.

| Component | Model | Purpose |
|---|---|---|
| VLM | Qwen3-VL-8B-Instruct | Local/Global Reasoner |
| ASR | Whisper-large-v3-turbo | Speech transcription |
| Speaker encoder | ECAPA-TDNN (SpeechBrain) | Embedding + aggregation |
| Speech separator | MossFormer2 (ModelScope) | Overlapping speech |
| Mask generator | SAM3-Video (Meta) | Instance mask generation |

### F.2. Hyperparameters and Hardware

The speaker similarity threshold is set to $\tau = 0.35$, the input frame rate is $r = 1$ FPS, and the IoU matching threshold for tracklet association is $0.3$. All experiments are run on a single 48 GB NVIDIA A6000 GPU. Table 6 sweeps the speaker similarity threshold $\tau$. The plateau region $\tau \in [0.30, 0.40]$ varies by only $0.56$ HOTA, with our default $\tau = 0.35$ sitting at its center. The value was fixed early in development using a quick scan on five videos and never re-tuned, indicating that AVTracker is robust to this hyperparameter.

## G. Local and Global Reasoner

The two VLM reasoners are driven by carefully designed prompts. We summarize their core specifications below; the complete prompt texts are released with the code.

*Table 6.* Sensitivity to the speaker similarity threshold $\tau$.

| $\tau$ | HOTA | DetA | AssA |
|---|---|---|---|
| 0.20 | 26.55 | 28.09 | 26.71 |
| 0.25 | 28.09 | 29.21 | 28.70 |
| 0.30 | 28.53 | 30.63 | 28.14 |
| 0.35 | 29.08 | 31.18 | 28.47 |
| 0.40 | 29.09 | 29.96 | 28.66 |
| 0.45 | 26.46 | 27.63 | 26.41 |
| 0.50 | 27.34 | 28.22 | 27.65 |

**Local Reasoner.** Given video frames and the corresponding speech transcript, the reasoner decides whether the speech originates from a visible person in the frame, conditioning on facial features, lip motion, hairstyle, and clothing. Candidate persons are marked with red bounding boxes. When a match exists, the reasoner emits the speaker's bounding box for each frame; otherwise it emits $[0, 0, 0, 0]$. The output is a JSON object with two fields: `rationale` (free-form reasoning) and `boxes` (per-frame coordinates).

**Global Reasoner.** Given persons across frames, the reasoner determines which instances belong to the same individual. It groups frames by identity and assigns each person a unique label (`person_0`, `person_1`, ...). The output is a JSON object with two fields: `rationale` and `persons`, where `persons` maps each `person_id` to its frame indices.

## H. Computational Complexity and Efficiency

AVTracker trades inference speed for accuracy by leveraging a VLM. Table 7 compares it against end-to-end baselines, and Table 8 attributes its cost to individual components.

*Table 7.* Parameter count, per-frame FLOPs, and throughput.

| Method | Type | Params | GFLOPs/frame | FPS |
|---|---|---|---|---|
| AVISM / ACVIS | End-to-end | 238M | ~308 | ~2–5 |
| AVTracker | Pipeline | ~9.4B | ~18,900 | 0.21 |

*Table 8.* Per-component FLOPs breakdown of AVTracker.

| Component | GFLOPs/frame | Proportion |
|---|---|---|
| Whisper | ~35 | <1% |
| SAM3-Video | ~630 | 3% |
| Qwen3-VL-8B-Instruct | ~18,200 | 97% |

Qwen3-VL dominates the cost. Despite being roughly $60\times$ more expensive than end-to-end baselines, AVTracker delivers substantially higher HOTA, suggesting that for the difficulty level of AVTrack the foundation-model orchestration paradigm is currently the more accuracy-efficient

choice. Quantization, batched API serving, and KV-cache reuse across the Local and Global Reasoners are natural avenues for reducing this overhead.

## I. Per-Challenge Performance Analysis

To localize where current methods struggle, we report HOTA broken down by the eight challenge categories of AVTrack in Table 9. The category proportions in AVTrack are: Camera Motion 90.5%, Visual Occlusion 80.9%, Position Change 70.7%, Background Switch 60.5%, Multi-turn Sounding 56.8%, Instance Scale Dynamics 56.9%, and Audio–Visual Inconsistency 9.2%. Each of these proportions far exceeds those in prior benchmarks such as AVISeg, where most categories appear in less than 10% of clips.

*Table 9.* Per-challenge HOTA comparison across all baselines.

| Challenge | AVISM | ACVIS | AVTrackFormer | AVTracker |
|---|---|---|---|---|
| Multiple Instances | 20.2 | 20.1 | 19.8 | **27.0** |
| Instance Scale | 18.6 | 18.2 | 17.8 | **26.8** |
| Visual Occlusion | 20.1 | 19.8 | 19.2 | **28.1** |
| Position Change | 20.6 | 20.7 | 20.7 | **25.9** |
| Camera Motion | 21.0 | 20.9 | 19.9 | **29.6** |
| Background Switch | 20.1 | 19.9 | 19.0 | **28.9** |
| Multi-turn Sound | 21.1 | 20.8 | 20.1 | **29.2** |
| AV Inconsistency | 12.6 | 12.8 | 12.2 | **18.5** |

**Failure modes.** Within AVTracker, the relatively weaker scenarios are Visual Occlusion (HOTA 28.1) and Multiple Instances (27.0), both well below the strongest Camera Motion case (29.6). The dominant failure pattern in these categories is tracklet association ambiguity when occluded speakers alternate at close range, leading the Global Reasoner to merge or split identities incorrectly. Even in these worst-case scenarios, AVTracker still leads the strongest end-to-end baseline by more than 7 HOTA, suggesting that the local-to-global formulation degrades gracefully under heavy occlusion rather than collapsing.

## J. Comparison with Commercial Omni-LLMs

As shown in Table 10, to position AVTrack against the rapidly improving frontier of commercial audio-visual LLMs, we run Gemini 2.5 Pro in a zero-shot regime: video frames at 1 FPS and the raw audio track are fed in directly (no transcript), and the model is prompted to output the active-speaker bounding box per frame together with a cross-frame identifier. Gemini 2.5 Pro reaches only HOTA = 14.4, below specifically trained end-to-end baselines ($\sim$20) and far below AVTracker (29.1). Despite their breadth on general audio–visual understanding tasks, current commercial AV LLMs still struggle to localize and track active speakers directly under the conditions assembled in AVTrack, which reinforces the value of AVTrack as a testbed for the next generation of these models.

*Table 10.* Comparison with commercial audio-visual LLMs.

| Method | Type | HOTA | DetA | AssA |
|---|---|---|---|---|
| AVISM | End-to-end trained | 20.8 | 23.2 | 19.5 |
| ACVIS | End-to-end trained | 20.6 | 22.6 | 19.7 |
| Gemini 2.5 Pro | AV LLM (zero-shot) | 14.4 | 13.8 | 15.9 |
| AVTracker | Pipeline (training-free) | 29.1 | 31.2 | 28.5 |

## K. Scope and Future Directions

### K.1. Human-centric Focus

AVTrack focuses on human-centric scenes for three main reasons. First, human understanding is central to many real-world applications, such as video conferencing, surveillance, accessibility, and human-computer interaction. Second, humans introduce challenges that differ from those in generic object tracking, including multi-turn speech, long-term speaker identity association, and subtle appearance differences between individuals. These factors make dedicated evaluation necessary. Third, AVTrack complements existing benchmarks such as AVISeg, which already addresses general audio-visual instance segmentation, by providing a benchmark specifically designed for human-centric scenarios.

### K.2. Future Work

Looking forward, we plan to extend AVTrack along three concrete directions. First, we will enlarge the annotations to support *human-centric reference audio-visual segmentation* (**Human-centric Ref-AVS**), in which models must localize and segment specific individuals based on natural language expressions in scenes with rich auditory and visual cues. This extension is intended to encourage multimodal reasoning that integrates auditory, visual, and linguistic information for precise human-centric scene understanding. Second, motivated by the per-challenge analysis in Section I, we target architectural improvements for the hardest categories, especially Audio-Visual Inconsistency and Instance Scale Dynamics, where all current methods plateau. Third, we will explore agentic reasoning mechanisms such as memory-augmented trackers and reflection-based error correction, which are well matched to the local-to-global formulation and could further unlock VLM reasoning in long and crowded sequences.

