# OpenReview forum: "AVTrack: Audio-Visual Tracking in Human-centric Complex Scenes"
_ICML.cc/2026/Conference — ICML 2026 regular_

### Official Review · Reviewer_nLU6 · 2026-02-25

**Soundness:** 3
**Presentation:** 3
**Significance:** 3
**Originality:** 3
**Overall Recommendation:** 4
**Confidence:** 4

**Summary:**

The paper introduces AVTrack, a human-centric audio-visual instance segmentation (AVIS) benchmark specifically curated to stress-test tracking in complex, dynamic, real-world scenes. AVTrack comprises 871 clips (≈54s on average) with 3,120 annotated instance tracklets and explicitly targets eight challenging conditions (e.g., camera motion, occlusion, relative position change, audio-visual inconsistency, multi-turn speaking). The authors benchmark VIS and AVIS methods and observe large performance drops relative to existing datasets, and they propose AVTracker, a modular, training-free, local-to-global pipeline that leverages foundation models (Whisper, ECAPA-TDNN, SAM3, Qwen3-VL, MossFormer2) to set a strong baseline.

**Compliance With Llm Reviewing Policy:**

Affirmed.

**Final Justification:**

The authors fully resolved my concerns and I keep my rating as weak accept.

**Key Questions For Authors:**

- Performance of other competitive methods on the challenges of AVTrack dataset mentioned on Fig.2?
- If other methods can receive a good performance, the contribution of the proposed AVTrack is limited.

**Limitations:**

yes

**Strengths And Weaknesses:**

- **Strengths**
    - The dataset is built on a clear taxonomy, effectively addressing human-centric, multi-person, and dynamic scenarios that are underrepresented in prior AVIS datasets.
    - The paper provides a baseline implementation with superior performance.

- **Weaknesses**
    - The proposed baseline is a pipeline of distinct MLLMs rather than a unified model. While effective, this limits its utility for the community. I strongly suggest providing an **end-to-end trainable baseline** to better support future research and fair comparison.
    - The specific prompts used for the "Local/Global Reasoner" modules are not disclosed, which hinders reproducibility.
    - While the dataset introduces complex scenarios (e.g., dynamic camera motion, multi-turn interactions), the paper lacks specific technical contributions to address these challenges effectively.

---

> ### Author Rebuttal · Authors · 2026-03-31
>
> # Response to Reviewer nLU6
>
> We thank the reviewer for the positive evaluation and constructive suggestions. We particularly appreciate the recognition that the AVTrack dataset is "built on a clear taxonomy" and that the baseline demonstrates "superior performance." We address each concern below.
>
> ---
>
> ## W1: Need for an end-to-end trainable baseline
>
> Thank you for this suggestion. In fact, we have already provided an end-to-end trainable baseline, AVTrackFormer, in Appendix D, which augments AVISM with bidirectional audio-visual fusion (AV-OFM) and achieves the best HOTA performance among all end-to-end methods. .
>
> In the revision, we will: (1) add explicit references to AVTrackFormer in the main experiment section; (2) emphasize that we provide both a trainable baseline (AVTrackFormer) and a training-free baseline (AVTracker).
>
> ---
>
> ## W2: Prompts for Local/Global Reasoner not disclosed
>
> Thank you for pointing this out. Below is a summary of the core prompt content:
>
> **Local Reasoner Prompt:** Given video frames and corresponding speech transcriptions, determine whether the speech originates from a visible person in the frame based on facial features, hairstyle, clothing, etc. Visible persons are annotated with red bounding boxes. If matched, output the speaker's bounding box for each frame; otherwise output [0,0,0,0]. Output format: JSON with "rationale" (reasoning) and "boxes" (per-frame bounding boxes).
>
> **Global Reasoner Prompt:** Analyze persons appearing across multiple frames and determine whether they are the same individual across frames. Group frames containing the same person and assign a unique identifier to each distinct individual (person_0, person_1, etc.). Output format: JSON with "rationale" and "persons" (person_id → frame_id list).
>
> The revised appendix will include the complete prompt text.
>
> ---
>
> ## W3: Lack of specific technical contributions addressing the challenges
>
> Thank you for this concern. Our core contribution is the benchmark itself, while AVTracker introduces several meaningful technical designs:
>
> 1. **Dynamic windowing strategy**: Unlike fixed windows, ASR-driven dynamic windows preserve complete semantic units, enabling more effective audio-visual association. Ablation experiments (Table 3, C2 vs. Base) show a contribution of +1.40 HOTA.
> 2. **Speaker chunks aggregation:** Cosine-similarity-based merging of adjacent chunks reduces redundancy while preserving speaker identity, significantly improving the scalability of global association (Table 3, C1 vs. M3: +7.13 HOTA).
> 3. **Local-to-global tracklet grouping:** The hierarchical association paradigm itself is a meaningful contribution to the AVIS field, demonstrating that structured reasoning outperforms end-to-end methods in overall.
>
> The revision will articulate these technical contributions more clearly.
>
> ---
>
> ## Q1: Performance of other methods on AVTrack challenges
>
> Thank you for this suggestion. The revised manuscript includes a per-challenge breakdown for all methods across the 8 challenge conditions:
>
> | Challenge | AVISM | ACVIS | AVTrackFormer | **AVTracker** |
> |-----------|-------|-------|---------------|-------------|
> | Multiple Instances | 20.2 | 20.1 | 19.8 | **27.0** |
> | Instance Scale | 18.6 | 18.2 | 17.8 | **26.8** |
> | Visual Occlusion | 20.1 | 19.8 | 19.2 | **28.1** |
> | Position Change | 20.6 | 20.7 | 20.7 | **25.9** |
> | Camera Motion | 21.0 | 20.9 | 19.9 | **29.6** |
> | Background Switch | 20.1 | 19.9 | 19.0 | **28.9** |
> | Multi-turn Sound | 21.1 | 20.8 | 20.1 | **29.2** |
> | AV Inconsistency | 12.6 | 12.8 | 12.2 | **18.5** |
>
> All existing methods perform poorly across all challenge conditions (HOTA < 21), further validating the value of AVTrack as a challenging benchmark.
>
> ---
>
> ## Q2: If other methods perform well, the contribution is limited
>
> As shown in Table 2 and per-challenge breakdown table above , no existing end-to-end method performs well on AVTrack and better than AVTracker. The best end-to-end model achieves only 21.47 HOTA, far below a satisfactory level. Even AVTracker (our strongest baseline) achieves only 29.08 HOTA. The per-challenge analysis above further confirms that all methods exhibit significant performance gaps across every challenge condition, validating that AVTrack exposes fundamental limitations of current methods.

---

> > ### Author Rebuttal · Reviewer_nLU6 · 2026-04-05
> >
> > The authors fully addressed my concerns.

---

> > > ### Author Response · Authors · 2026-04-05
> > >
> > > We sincerely appreciate the reviewer's constructive suggestions and will incorporate the suggested content and experiments in the revised version.

---

### Official Review · Reviewer_cM2q · 2026-03-12

**Soundness:** 2
**Presentation:** 2
**Significance:** 2
**Originality:** 2
**Overall Recommendation:** 4
**Confidence:** 3

**Summary:**

This paper introduces AVTrack, a human-centric audio-visual instance segmentation dataset, along with a simple baseline, AVTracker. Existing datasets are limited in evaluating realistic scenarios with dynamic camera motion, frequent occlusion, multi-person interaction, and complex spatiotemporal dynamics. AVTrack helps address this gap by collecting more diverse and challenging scenes. Experiments show that existing models degrade notably on AVTrack, while the proposed baseline brings some improvement and provides a reasonable starting point.

**Compliance With Llm Reviewing Policy:**

Affirmed.

**Final Justification:**

All the concerns are clearly resolved.

**Key Questions For Authors:**

See weaknesses

**Limitations:**

Yes

**Strengths And Weaknesses:**

## Strength
- Proposing an audio-visual instance segmentation dataset for complex real-world scenes is a valuable contribution.
- The modular baseline is practical and can be improved by replacing individual components.


## Weakness
### Limited novelty in the baseline
- The baseline is presented as a simple modular framework, which is acceptable, but some parts of the pipeline remain unclear.
- The speech separation module is not sufficiently explained. In overlapping speech cases, it is unclear whether the number of speakers is estimated automatically or assumed beforehand. Also, a^hat was introduced in line 268, but the later equation in Eq.1 seems to use a instead, so its actual role in the pipeline is unclear.
- In Section 4.3, the transcript is used to localize the active speaker using VLM. However, in real scenarios, the transcript may correspond to narration or off-screen speech, so the method may rely on semantic context rather than true audio-visual correspondence.

### Limited experimental analysis
- More analysis is needed on which scenarios are particularly challenging, to better show both the dataset difficulty and the model limitations.
- In Figure 6, AVTracker seems to fail not only in multi-speaker tracking but sometimes in tracking any speaker reliably. This failure case should be discussed more clearly.

### Limited dataset analysis
- More detailed statistics would be helpful, such as source distribution, scenario types, number of overlapping speakers, and degree of occlusion.

---

> ### Author Rebuttal · Authors · 2026-03-31
>
> # Response to Reviewer cM2q
>
> We thank the reviewer for the detailed feedback and recognition of our dataset contribution and modular baseline design. We address each concern below.
>
> ---
>
> ## W1: Parts of the Baseline Pipeline Remain Unclear
>
> Thanks for pointing out this question.
>
> AVTracker's three-stage pipeline: Stage 1: Whisper ASR generates transcription + ECAPA-TDNN speaker embeddings aggregate speech segments by speaker identity; Stage 2: Qwen3-VL Local Reasoner identifies per-frame speaker bboxes + SAM3 generates masks, then forming local tracklets; Stage 3: Qwen3-VL Global Reasoner groups cross-frame identities based on visual appearance of keyframes, then merging into global trajectories.
>
> The revised manuscript will include detailed descriptions of each stage, complete Local/Global Reasoner prompts and explicit data flow between modules.
>
> ---
>
> ## W2: Insufficient Explanation of Speech Separation; Notation Inconsistency
>
> We thank the reviewer for noting this. The revised version uniformly uses hat_a for separated enhanced speech signals.
>
> The number of speakers is not automatically estimated. MossFormer2 as a source separation model decomposes the mixture into independent sources; the speaker embedding encoder (ECAPA-TDNN) then matches each source to the current speaker chunk via cosine similarity. When no overlap is detected, separation is bypassed.
>
> Table 3 ablation provides objective data:
> - Base (no separation): HOTA=28.85, AssA=27.39
> - +MossFormer2: HOTA=29.08, AssA=28.47
> - +SepFormer: HOTA=28.41, showing unreliable separation introduces noise
>
> ---
>
> ## W3: Reliance on Semantic Context (Transcription)
>
> We thank the reviewer for this insightful observation. The transcript serves as a semantic bridge between modalities, not the sole localization basis. The Local Reasoner jointly conditions on transcript and visual context of SAM3-detected candidates to determine the active speaker. The transcript disambiguates (e.g., when multiple people are visible), but final association is grounded in visual evidence.
>
> For narration/off-screen speech ("Audio-Visual Inconsistency", 9.2% of AVTrack), per-challenge analysis shows this is the hardest scenario for all methods:
>
> | Method | AV Inconsistency HOTA |
> |-|-|
> | AVISM | 12.6 |
> | ACVIS | 12.8 |
> | AVTrackFormer | 12.2 |
> | **AVTracker** | **18.5** |
>
> AVTracker still leads, showing the VLM can partially recognize unmatched speech. Transcription is obtainable via ASR (e.g., Whisper), standard in video conferencing and captioning. We will adds off-screen speech discussion and joint conditioning clarification in the revised version.
>
> ---
>
> ## W4: Limited Experimental Analysis
>
> We appreciate this suggestion. The revision includes per-challenge HOTA breakdown for all methods:
>
> | Challenge | AVISM | ACVIS | AVTrackFormer | **AVTracker** |
> |-----------|-------|-------|---------------|-------------|
> | Multiple Instances | 20.2 | 20.1 | 19.8 | **27.0** |
> | Instance Scale | 18.6 | 18.2 | 17.8 | **26.8** |
> | Visual Occlusion | 20.1 | 19.8 | 19.2 | **28.1** |
> | Position Change | 20.6 | 20.7 | 20.7 | **25.9** |
> | Camera Motion | 21.0 | 20.9 | 19.9 | **29.6** |
> | Background Switch | 20.1 | 19.9 | 19.0 | **28.9** |
> | Multi-turn Sound | 21.1 | 20.8 | 20.1 | **29.2** |
> | AV Inconsistency| 12.6 | 12.8 | 12.2| **18.5** |
>
> The above table shows that: (1) AVTracker leads by 6-10 HOTA across all 8 scenarios; (2) AV Inconsistency is hardest (HOTA 12-18) subset compared with others; (3) end-to-end methods show similar performance across scenarios, indicating fundamental capability bottlenecks.
>
> ---
>
> ## W5: Tracking Failures in Figure 6
>
> We thank the reviewer for this point. Per-challenge analysis pinpoints AVTracker's weaker scenarios: Visual Occlusion (HOTA 28.1) and Multiple Instances (27.0) vs. Camera Motion (29.6). The main cause is tracklet association ambiguity when occluded speakers alternate at close range. Even here, AVTracker leads end-to-end methods substantially, showing the local-to-global paradigm's robustness. The revision will link failure modes to specific challenge categories and give more detailed descriptions in detail.
>
> ---
>
> ## W6: Limited Dataset Analysis
>
> We appreciate this suggestion. Section 3.2 and Figures 2 show source distribution and challenge proportions: Camera Motion 90.5%, Visual Occlusion 80.9%, Position Change 70.7%, Background Switch 60.5%, Multi-turn Sounding 56.8%, Instance Scale 56.9%, AV Inconsistency 9.2%, far exceeding AVISeg (most <10%). In the revised version, we will provide more a clearer description explicitly.

---

> > ### Author Rebuttal · Reviewer_cM2q · 2026-04-03
> >
> > Thanks for the effort on making all the things clear.

---

> > > ### Author Response · Authors · 2026-04-03
> > >
> > > We sincerely appreciate your time and the constructive feedback you have provided. We commit to incorporating additional experimental results and improving the overall readability of the manuscript in the revised version.

---

### Official Review · Reviewer_V6aj · 2026-03-12

**Soundness:** 3
**Presentation:** 3
**Significance:** 4
**Originality:** 3
**Overall Recommendation:** 4
**Confidence:** 5

**Summary:**

This paper focuses on the human-centric audio-visual instance segmentation (AVIS) task, aiming to address the issue that existing benchmarks are overly simplified for static scenes and thus fail to strictly evaluate the robust spatio-temporal modeling and cross-modal reasoning capabilities of models in dynamic real-world environments. The authors propose a new human-centric benchmark dataset, AVTrack, which contains 871 clips and eight challenging scenarios such as occlusion and audio-visual inconsistency. Additionally, the authors introduce AVTracker, which consists of three modules: speaker clip aggregation, local window processing, and global window processing. AVTracker combines pre-trained audio-visual features with a local-to-global association mechanism to robustly associate instance trajectory clips over time, thereby addressing the challenges of complex AVIS tasks. Comprehensive experiments on AVTrack show that the performance of the current state-of-the-art video instance segmentation and AVIS methods significantly drops in this benchmark, while AVTracker outperforms all existing baselines by a large margin.

**Compliance With Llm Reviewing Policy:**

Affirmed.

**Final Justification:**

All my concerns are addressed, so I maintain my score.

**Key Questions For Authors:**

- Computational complexity and practical deployment. Could you report the computational complexity (FLOPs) and inference speed (FPS) of AVTracker and its competing baselines on a standard hardware configuration?
-Performance and hyperparameter sensitivity for each challenge. Could you provide a performance breakdown for each of the 8 challenging conditions in AVTrack for AVTracker and all baselines, particularly the HOTA, DetA, and AssA scores?

**Limitations:**

yes

**Strengths And Weaknesses:**

- Strength

The AVTrack benchmark is carefully built, with clear scenario definitions, a reliable annotation pipeline, and strong quality control. The experiments are fairly complete, and the AVTracker design is sensible for the task.
The presentation is also good. The paper is easy to follow, and the figures and tables do a good job of showing the main ideas and results.
The main value of the work is that it provides a more realistic benchmark for human-centric audio-visual instance segmentation and shows that current methods still struggle in complex settings. The benchmark setup and the tracking framework also give the paper a reasonable level of novelty.

- Weakness

The paper has several weaknesses. Important experimental details are also missing, including runtime, complexity, parameter sensitivity, and objective evidence for the effect of speech separation.
The presentation is acceptable, but reproducibility is limited by missing implementation details. The descriptions of the local and global reasoners are also too vague, and the discussion of future work remains rather general.
The benchmark is useful but still narrow in scope, and the method mainly relies on integrating existing pre-trained models rather than introducing a clearly new technical component.

Overall, it is still a good benchmark for this field.

---

> ### Author Rebuttal · Authors · 2026-03-31
>
> # Response to Reviewer V6aj
>
> We sincerely thank the reviewer for the thorough review. We particularly appreciate the positive assessment of AVTrack and the recognition of the benchmark's coverage and comprehensiveness. We address each concern below.
>
> ---
>
> ## W1: Missing Runtime, Computational Complexity, and Parameter Sensitivity
>
> We thank the reviewer for pointing out this important gap. The revised manuscript will include the following information:
>
> ### Computational Complexity and Inference Speed
>
> |Method|Type|Params| GFLOPs/frame|FPS|
> |-|-|-|-|-|
> |AVISM/ACVIS|E2E|238M|~308|~2-5|
> |AVTracker|Pipeline|~9.4B|~18,900|0.21|
>
> Per-component FLOPs breakdown of AVTracker:
>
> |Component|GFLOPs/frame|Proportion|
> |-|-|-|
> | Whisper|~35|<1%|
> | SAM3-Video|~630|3%|
> | Qwen3-VL-8B-Instruct|~18,200|97%|
>
> Qwen3-VL-8B-Instruct is the dominant computational bottleneck. Despite ~60× higher FLOPs than E2E methods, AVTracker achieves substantially superior HOTA, demonstrating the performance advantage of the foundation model orchestration paradigm for complex AVIS tasks. Future work can significantly reduce overhead through quantization or API-level parallelism.
>
> ### Parameter Sensitivity Analysis (Threshold τ)
>
> We conducted a sweep analysis on the speaker similarity threshold τ:
>
> |τ | HOTA | DetA | AssA |
> |-|-|-|-|
> |0.20|26.55| 28.09 | 26.71|
> |0.25|28.09| 29.21 | 28.70|
> |0.30|28.53| 30.63 | 28.14|
> |0.35|29.08| 31.18 | 28.47|
> |0.40|29.09| 29.96 | 28.66|
> |0.45|26.46| 27.63 | 26.41|
> |0.50|27.34| 28.22 | 27.65|
>
> τ=0.30–0.40 is the optimal range, with HOTA varying by only 0.56 points and τ=0.35 at the center. This value was determined early in the project through a quick experiment on 5 videos without fine-grained tuning, indicating that AVTracker is insensitive to this hyperparameter.
>
> ---
>
> ## W2: Missing Implementation Details / Reproducibility
>
> We thank the reviewer for pointing this out. Key implementation specifications are as follows:
>
> |Component|Model|Purpose|
> |-|-|-|
> |VLM |Qwen3-VL-8B-Instruct|Local/Global Reasoner|
> |ASR |Whisper-large-v3-turbo|Speech transcription|
> |Speaker|ECAPA-TDNN|Speaker embedding + aggregation|
> |Separator|MossFormer2|Overlapping speech separation|
> |Mask|SAM3-Video|Instance mask generation|
>
> Key hyperparameters: τ=0.35, frame rate=1 FPS, IoU threshold=0.3, running on 48GB A6000. The revised manuscript will include the above details, and the full prompts and concrete reasoning examples for the Local/Global Reasoners will be provided in the appendix.
>
> ---
>
> ## W3: Future Work Discussion Too General
>
> We appreciate this suggestion. The revised manuscript will make the future directions more specific:
> 1. Proposing concrete architectural improvements targeting the most challenging conditions identified by per-challenge analysis (AV Inconsistency, Instance Scale Dynamics)
> 2. Exploring agentic reasoning mechanisms (e.g., memory-augmented trackers, reflection-based error correction) to further unlock VLM reasoning potential in complex scenarios
>
> ---
>
> ## W4: Benchmark Narrow in Scope (Human-Centric Only)
>
> The human-centric focus is a deliberate design choice:
>
> 1. Practical importance: Human-centric understanding is fundamental to core application scenarios such as video conferencing, surveillance, accessibility, and human-computer interaction
> 2. Unique challenges: Multi-turn speech, speaker identity maintenance, and fine-grained appearance similarity between individuals are qualitatively different from general object tracking
> 3. Complementary to existing benchmarks: AVISeg already covers general audio-visual instance segmentation; AVTrack complements it by providing a specialized, human-centric test platform
>
> ---
>
> ## W5: Method Mainly Relies on Existing Pretrained Models
>
> Using mature pretrained models ensures reproducibility and provides a fair baseline for the community. The modular design allows researchers to isolate the impact of individual components for targeted improvements. AVTracker's HOTA vs. the best e2e method demonstrates that the system orchestration approach itself carries significant value, serving as an important baseline and prototype for future agentic reasoning research in this domain.
>
> ---
>
> ## Q1: Per-Challenge Performance Breakdown
>
> |Challenge|AVISM|ACVIS|AVTrackFormer|AVTracker|
> |-|-|-|-|-|
> |Multi. Inst.|20.2|20.1|19.8|27.0|
> |Inst. Scale|18.6|18.2|17.8|26.8|
> |Vis. Occ.|20.1|19.8|19.2|28.1|
> |Position Change|20.6|20.7|20.7|25.9|
> |Camera Motion |21.0|20.9|19.9|29.6|
> |Background Switch|20.1|19.9|19.0|28.9|
> |Multi-turn Sound|21.1|20.8|20.1|29.2|
> |AV Incon.|12.6|12.8|12.2|18.5|
>
> AVTracker consistently leads by 6–10 HOTA across all conditions. E2e methods show similar performance across scenarios, suggesting that the bottleneck lies in fundamental capability rather than specific scenarios. AV Inconsistency is the most challenging for all methods, revealing a key research direction. Due to space constraints, complete per-challenge HOTA/DetA/AssA scores will be provided in the revised manuscript.

---

> > ### Author Rebuttal · Reviewer_V6aj · 2026-04-02
> >
> > All my concerns are addressed.

---

> > > ### Author Response · Authors · 2026-04-02
> > >
> > > Thank you sincerely again for your thorough and constructive review! We are glad that all your concerns have been addressed and truly appreciate the time and effort you devoted to evaluating our manuscript.

---

### Official Review · Reviewer_qRw9 · 2026-03-13

**Soundness:** 3
**Presentation:** 2
**Significance:** 3
**Originality:** 3
**Overall Recommendation:** 5
**Confidence:** 3

**Summary:**

This paper addresses the complex task of tracking human using audio-video knowledge in real-world setting and long duration. The paper proposes a dataset and a simple effective baseline.

**Compliance With Llm Reviewing Policy:**

Affirmed.

**Final Justification:**

The authors have provided detailed responses and additional results to address my main concern. Thus I updated my score to 5.

**Key Questions For Authors:**

1. Will the AV Tracker work if the transcription is not present? How will you adjust the algorithm to handle this scenario.
2. Which version and size of LLM Ami’s being used? Can you justify your design choices?
3. Please provide more details on the experimental setup.
4. How does proprietary AV LLM work in this scenario?

**Limitations:**

Yes

**Strengths And Weaknesses:**

Strength:
1. The proposed AVTrack dataset seems solid and can significantly impact the progress in the field

Weakness:
1. The local window processing stage relies on transcription. Where will we get transcription in real-world?
2. Which version of the LLM is used in AVTracker? There is no mention beside the icon in Fig 4.
3. The experimental details are non-existant.
4. How to make the tracker work without transcription? Did you explore the scope of speaker recognition for tracking? This seems like one of the feasible approach with audio-video.

---

> ### Author Rebuttal · Authors · 2026-03-31
>
> # Response to Reviewer qRw9
>
> We thank the reviewer for the positive evaluation. We especially appreciate the reviewer's recognition that the AVTrack dataset "seems solid and can significantly impact the progress in the field." Below we address the four core themes raised by the reviewer.
>
> ---
>
> ## 1. Role of Transcription (W1, W4, Q1)
>
> The transcription in AVTracker is fully automatically generated from raw audio by Whisper-large-v3-turbo, requiring no manual input. Whisper has been widely deployed in production environments such as video conferencing (Zoom, Teams) and captioning systems (YouTube), and is a mature industrial-grade component. For noisy and overlapping speech scenarios, the MossFormer2 module in the pipeline performs speech separation preprocessing before Whisper.
>
> Regarding speaker recognition: AVTracker's Stage 1 uses the ECAPA-TDNN speaker encoder to extract speaker embeddings and aggregates speech segments belonging to the same speaker into speaker chunks via cosine similarity (threshold τ=0.35). This step effectively reduces the burden on the subsequent Global Window Process. The ablation study in Table 3 (C1 vs M3) shows that removing chunk compression causes HOTA to drop sharply from 24.01 to 16.88, demonstrating the critical role of speaker embedding-driven aggregation for overall performance.
>
> On this basis, the role of transcription is semantic disambiguation: when multiple visually similar individuals are present in the scene, speech content provides the VLM with additional discriminative cues, helping the Local Reasoner identify the current speaker.
>
> Regarding approaches that do not rely on transcription: we provide **AVTrackFormer** in Appendix D, an end-to-end baseline trained on AVISM that directly learns cross-modal associations from raw audio-visual signals, requiring neither ASR nor transcription. AVTrackFormer achieves the best HOTA among end-to-end methods (21.47, Table 2). The two baselines represent complementary technical directions: AVTracker achieves strong performance through foundation model orchestration + semantic disambiguation (HOTA 29.08), while AVTrackFormer provides an end-to-end trainable approach that does not rely on transcription.
>
> ---
>
> ## 2. VLM Version and Design Rationale (W2, Q2)
>
> We thank the reviewer for this important suggestion. Section 4.1 (Framework Overview) of the paper states that Qwen3-VL is used as the VLM backbone; the missing specific model variant is **Qwen3-VL-8B-Instruct**. The rationale for this choice:
>
> - Qwen3-VL has gained widespread recognition in the vision-language understanding community, particularly excelling in fine-grained spatial reasoning and multi-image understanding, making it well-suited for the joint visual-textual reasoning tasks required by AVTracker.
> - The 8B scale balances performance and resource consumption. The ablation study (Table 3, M2 vs Base) shows that reducing from 8B to 4B leads to a HOTA decrease (28.85→24.47), confirming the importance of model capacity for audio-visual reasoning quality.
>
> We will add the specific model variant in the method section of the revised version and supplement the legend in Figure 4.
>
> ---
>
> ## 3. Experiments and Implementation Details (W3, Q3)
>
> We thank the reviewer for pointing out the missing experimental details. Below is the complete model and hyperparameter configuration:
>
> | Component | Model |
> |-----------|-------|
> | VLM | Qwen3-VL-8B-Instruct
> | ASR | Whisper-large-v3-turbo
> | Speaker Embedding | ECAPA-TDNN (speechbrain)
> | Speech Separation | MossFormer2 (ModelScope)
> | Mask Generation | SAM3-Video (facebook)
>
> Key hyperparameters: speaker similarity threshold τ=0.35, frame rate r=1 FPS, IoU matching threshold=0.3.
>
> Due to the response length limitation, we will supplement the complete prompts for the Local and Global Reasoner in the revised appendix. We appreciate your understanding.
>
> ---
>
> ## 4. Commercial AV LLM Evaluation (Q4)
>
> We thank the reviewer for raising this cutting-edge question. To directly address it, we conducted an end-to-end AV LLM experiment using Gemini 2.5 Pro: inputting video frames (1 FPS) and raw audio (without transcript), zero-shot directly outputting the bounding box of the active speaker per frame and cross-frame identity. The results show that Gemini 2.5 Pro achieves HOTA=14.4, lower than specifically trained end-to-end methods (~20) and far below AVTracker (29.1):
>
> | Method | Type | HOTA | DetA | AssA |
> |--------|------|------|------|------|
> | AVISM | End-to-end trained | 20.8 | 23.2 | 19.5 |
> | ACVIS | End-to-end trained | 20.6 | 22.6 | 19.7 |
> | Gemini 2.5 Pro | AV LLM (zero-shot) | 14.4 | 13.8 | 15.9 |
> | AVTracker | Pipeline (training-free) | 29.1 | 31.2 | 28.5 |
>
> This result indicates that, despite excelling at general understanding tasks, current commercial AV LLMs still struggle to directly handle complex AVIS tasks. AVTrack provides a valuable testbed for evaluating the future progress of these models.

---

> > ### Author Rebuttal · Reviewer_qRw9 · 2026-03-31
> >
> > I thank the authors for providing the detailed rebuttal and for performing the additional evaluation that addresses my concerns. With the promised details about experimental setup along with the new results and explanations, I am happy change my score to 5.

---

> > > ### Author Response · Authors · 2026-04-01
> > >
> > > We sincerely thank the reviewer for the positive reassessment and the constructive questions that helped us further strengthen the manuscript.
> > >
> > > All promised revisions (including complete implementation details, full prompt specifications, and the new proprietary VLM baseline results) will be incorporated into the revised version.
> > >
> > > We are also committed to maintaining AVTrack as a long-term, open benchmark for human-centric audio-visual tracking, with continued updates on the latest progress to serve the community.

---

### Decision · Program_Chairs · 2026-04-30

**Decision:**

Accept (regular)

**Comment:**

Reviewers have achieved agreement on accepting this paper.